# Intestinal epithelium-derived BMP controls stem cell self-renewal in *Drosophila* adult midgut

**Aiguo Tian[1], Jin Jiang[1,2]\***

[1]Department of Developmental Biology, University of Texas Southwestern Medical Center at Dallas, Dallas, United States; [2]Department of Pharmacology, University of Texas Southwestern Medical Center at Dallas, Dallas, United States

**Abstract** Stem cells are maintained in a specialized microenvironment called niche but the nature of stem cell niche remains poorly defined in many systems. Here we demonstrate that intestinal epithelium-derived BMP serves as a niche signal for intestinal stem cell (ISC) self-renewal in *Drosophila* adult midgut. We find that BMP signaling is asymmetric between ISC and its differentiated daughter cell. Two BMP ligands, Dpp and Gbb, are produced by enterocytes and act in conjunction to promote ISC self-renewal by antagonizing Notch signaling. Furthermore, the basement membrane-associated type IV collagens regulate ISC self-renewal by confining higher BMP signaling to ISCs. The employment of gut epithelia as a niche for stem cell self-renewal may provide a mechanism for direct communication between the niche and the environment, allowing niche signal production and stem cell number to be fine-tuned in response to various physiological and pathological stimuli.

## Introduction

In adult life, many organs rely on stem cells to maintain their integrity by replenishing lost cells during tissue homeostasis and regeneration, yet the regulatory mechanisms that control stem cell proliferation, self-renewal, and differentiation are still not fully understood. Stem cells are thought to reside in specialized microenvironments called niches that produce signals required for stem cell self-renewal and maintenance (*Jones and Wagers, 2008*; *Morrison and Spradling, 2008*; *Losick et al., 2011*); however, the cellular basis of stem cell niches and the molecular nature of the niche signals have not been well defined in many systems.

*Drosophila* adult midgut has emerged as an attractive system to study stem cell biology in adult tissue homeostasis and regeneration not only because the cell lineage of this tissue is relatively simple and well defined but also because it bears similarities to the mammalian intestine (*Casali and Batlle, 2009*; *Biteau et al., 2011*; *Jiang and Edgar, 2012*). *Drosophila* posterior midgut contains self-renewing stem cells located adjacent to the basement membrane (BM) of the midgut epithelium (*Figure 1A*; *Micchelli and Perrimon, 2006*; *Ohlstein and Spradling, 2006*). These intestine stem cells (ISCs) undergo cell division and asymmetric fate determination to produce a renewed ISC and an enteroblast (EB). The EB exits cell cycle and differentiates into either an absorptive enterocyte (EC) or a secretory enteroendocrine cell (EE) depending on Notch (N) pathway activity (*Figure 1A*; *Ohlstein and Spradling, 2007*). Fate determination between the two ISC daughter cells is regulated by N signaling (*Micchelli and Perrimon, 2006*; *Ohlstein and Spradling, 2006*, *2007*; *Bardin et al., 2010*). Immediately after an ISC division, a high level of active Delta (Dl) is retained in the basally localized daughter cell that remains as ISC while the more apically localized daughter cell activates N signaling to become EB (*Ohlstein and Spradling, 2007*). How asymmetric N signaling between two ISC daughter cells is

**\*For correspondence:** jin.jiang@utsouthwestern.edu

**Competing interests:** The authors declare that no competing interests exist.

**Reviewing editor**: Utpal Banerjee, University of California, Los Angeles, United States

**eLife digest** Keeping an organ in top condition requires a steady supply of fresh cells to replace those that are dead or damaged. This is particularly critical for the epithelial cells lining the intestines, which only live for a few days, but are necessary for digesting food. These cells cannot simply reproduce by cell division, so they must be replenished by adult stem cells—adaptable cells that can produce any of the cell types found in a given organ.

When an adult stem cell divides, two daughter cells are produced. Normally, one of these remains in the stem state, and the other becomes a particular type of cell for use in the organ. Exactly how each daughter cell knows what to become is unclear. However, it is known that in addition to communicating with each other, stem cells also communicate with their immediate surroundings, which is known as a niche. For many processes, the molecules and mechanisms used in niche signaling remain to be discovered.

The midgut of fruit flies presents a relatively simple stem cell system for study, and has the added advantage that its cells behave in ways that are similar to the cells that make up the intestines of mammals. By developing a method of tracking the two daughter cells of a single stem cell simultaneously, Tian and Jiang have been able to uncover new details about how this niche operates.

Epithelial cells in the gut produce molecules called bone morphogenetic proteins (BMPs) that influence how bone and many other types of body tissues form. Tian and Jiang have found that two types of BMP are the signals responsible for keeping daughter cells in the stem state. When released from the base of the epithelial cells, BMPs can only travel a very short distance before other proteins trap them. As a result, one of a pair of daughter cells receives a higher level of the signal and remains as a stem cell. This cell then sends a signal to the other daughter cell, telling it to form a specialized cell.

established has remained poorly understood. A recent study suggested that asymmetric segregation of aPKC could play a role (*Goulas et al., 2012*), but additional mechanisms may exist. A previous study suggested that visceral muscle (VM)-derived Wingless (Wg) serves as a niche signal for ISC self-renewal (*Lin et al., 2008*). However, other studies suggested that Wg does not regulate ISC self-renewal but instead regulates its proliferation (*Lee et al., 2009*; *Cordero et al., 2012*). Hence, it is still unclear whether ISC fate is influenced by an environmental signal(s).

*Drosophila* midguts constantly undergo turnover and can regenerate after tissue damage (*Amcheslavsky et al., 2009*; *Jiang et al., 2009*). Several evolutionarily conserved signaling pathways, including Insulin, JNK, JAK-STAT, EGFR, Wg/Wnt, and Hpo pathways, have been implicated in the regulation of ISC proliferation during midgut homeostasis and regeneration (*Amcheslavsky et al., 2009*; *Buchon et al., 2009*; *Jiang et al., 2009*; *Lee et al., 2009*; *Karpowicz et al., 2010*; *Ren et al., 2010*; *Shaw et al., 2010*; *Staley and Irvine, 2010*; *Amcheslavsky et al., 2011*; *Biteau and Jasper, 2011*; *Jiang et al., 2011*; *Xu et al., 2011*; *Cordero et al., 2012*). It is very likely that additional pathways are involved in the regulation of midgut homeostasis and regeneration. By carrying out in vivo RNAi screen, we identified components in the BMP pathway as essential regulators of midgut regeneration. Clonal analysis and lineage tracing experiments suggest that BMP signaling regulates ISC self-renewal as well as ISC proliferation and lineage differentiation. We showed that EC-derived Dpp and Gbb act in concert to promote ISC self-renewal by antagonizing N signaling-mediated differentiation. We provided evidence that BMP exists in an apical-basal activity gradient and that BM regulates ISC self-renewal by confining high BMP signaling to ISCs.

## Results

### BMP signaling is required for *Drosophila* midgut regeneration

To identify additional genes and pathways that regulate injury-induced ISC proliferation, we carried out in vivo RNAi screen in which candidate genes were knocked down in midgut precursor cells using the *esg-Gal4 tub-Gal80^ts* (*esg^ts*) system, in which Gal4 is under the control of a temperature sensitive Gal80 (*McGuire et al., 2004*). 3–5-day-old adult females expressing individual *UAS-RNAi* transgenes

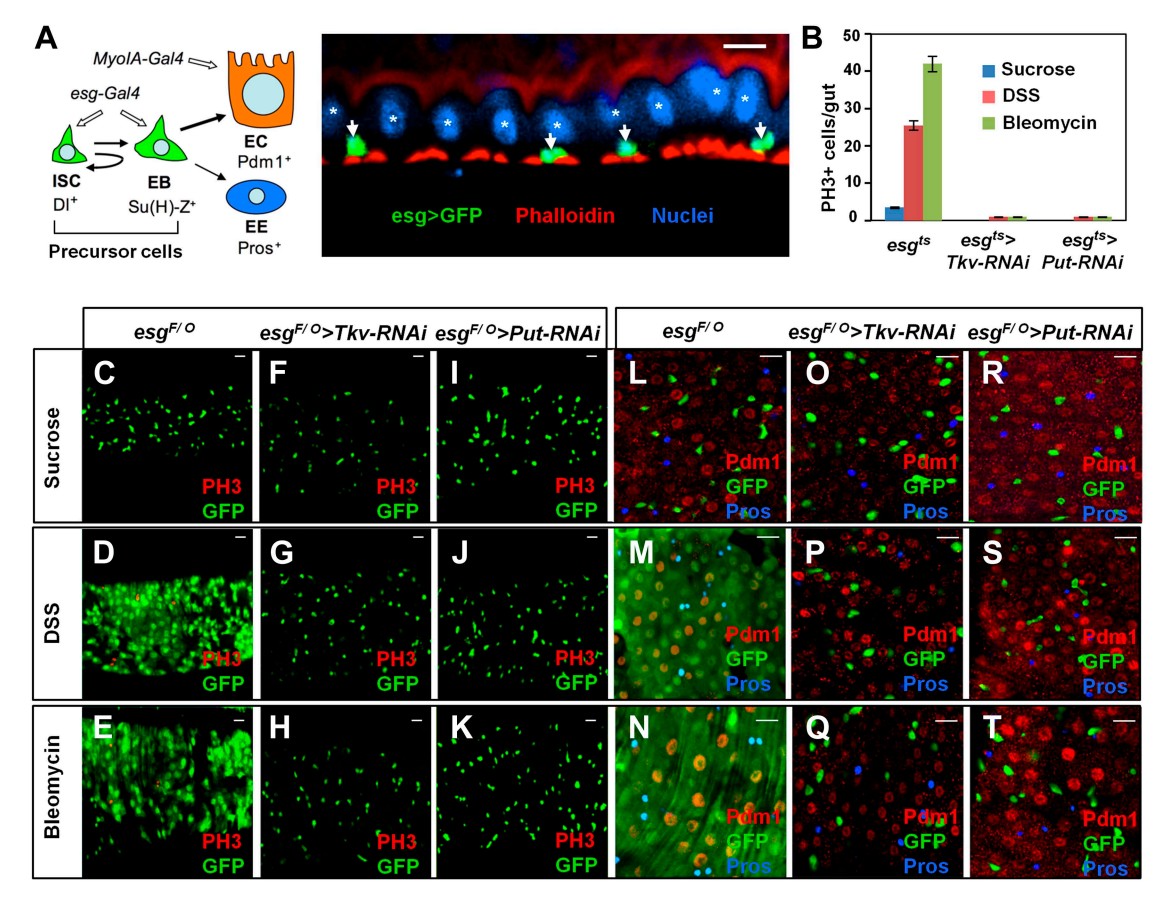

**Figure 1**. BMP signaling is required for midgut regeneration. (**A**) Left: an ISC lineage in *Drosophila* adult midguts. ISC: intestinal stem cell; EB: enteroblast; EC: enterocyte; EE: enteroendocrine cell. ISC and EB are collectively called precursor cells. Dl and Su(H)-lacZ mark ISC and EB, respectively, whereas Pdm1 and Pros are the markers for EC and EE, respectively. *esg-Gal4* and *Myo1A-Gal4* are precursor and EC-specific Gal4 drivers, respectively. Right: sagittal view of *Drosophila* midgut epithelium immunostained with an anti-GFP antibody (green), Phalloidin (red) and a nuclear dye (DRAQ5, blue). Arrows and asterisks indicate precursor cells and ECs, respectively. (**B**) Quantification of PH3+ cells in midguts from adults of the indicated genotypes (mean ± SD, n = 20 for each genotype). Tkv and Put RNAi in precursor cells blocked damage-induced mitotic index. (**C–T**) 3- to 5-day-old adult females of *esg^{F/O}* without (**C–E** and **L–N**) or with *UAS-Tkv-RNAi*[105834] (**F–H** and **O–Q**) or *UAS-Put-RNAi* (**I–K** and **R–T**) were shifted to 29°C for 8 days and treated with sucrose, DSS and bleomycin for 2 days, followed by immunostaining for GFP and PH3 (**C–K**), or GFP, Pdm1 and Pros (**L–T**). Top views of midguts are shown in these panels and in panels of all other figures unless indicated otherwise. Scale bars in this and other figures (except for *Figure 6A–C*) are 10 μm. *esg^{ts}*: *esg-Gal4 tub-Gal80^{ts}*. *esg^{F/O}*: *esg-Gal4 tub-Gal80^{ts} UAS-GFP; UAS-flp Act>CD2>Gal4*.

under the control of *esg^{ts}* were shifted to 29°C for 8 days and fed with tissue-damaging reagents such as DSS or bleomycin for 2 days, followed by immunostaining to examine ISC proliferation (*Ren et al., 2010*; *Amcheslavsky et al., 2011*; *Ren et al., 2013*). The TGFβ/BMP signaling pathway has been implicated as an important regulator of stem cell biology in many systems (*Zhang and Li, 2005*; *Oshimori and Fuchs, 2012*). In *Drosophila*, BMP signal is transduced via two type-I receptors Thickvein (Tkv) and Saxophone (Sax), and a type-II receptor Punt (Put) (*Nellen et al., 1994*; *Moustakas and Heldin, 2009*). We found that inactivation of BMP signaling in adult midgut precursor cells by knocking down either type I (*esg^{ts}>Tkv-RNAi*; VDRC#105834) or type II (*esg^{ts}>Put-RNAi*; VDRC #107071) receptor blocked DSS- or bleomycin-induced ISC proliferation, as indicated by the diminished mitotic cells recognized by staining with an anti-phospho-histone 3 (PH3) antibody (*Figure 1B*). This is somewhat surprising given that BMP signaling restricts stem cell/progenitor cell proliferation in mammalian intestines (*Haramis et al., 2004*; *He et al., 2004*).

To examine the role of BMP signaling in midgut regeneration, we employed the *esg^{F/O}* (*esg-Gal4 tub-Gal80^{ts} UAS-GFP; UAS-flp Act>CD2>Gal4*) system in which all cells in the ISC lineage are labeled

by GFP after shifting temperature to 29°C (*Jiang et al., 2009*). Feeding adult flies with DSS or bleomycin induced a rapid turnover of midgut epithelia, as evident by the newly formed ECs (marked by GFP⁺ Pdm1⁺) and EEs (GFP⁺, Pros⁺) 2–3 days after treatment (*Figure 1M,N*). These guts also contained many dividing ISCs marked by PH3 staining (*Figure 1D,E*). By contrast, mock treated guts only contained GFP⁺ precursor cells at this stage (*Figure 1C,L*). Damage-induced ISC proliferation and epithelial turnover were blocked by inactivation of either type I or type II BMP receptor because GFP⁺ ECs or EEs, or PH3⁺ cells were rarely found in midguts expressing *UAS-Tkv-RNAi¹⁰⁵⁸³⁴* or *UAS-Put-RNAi* with *esgᶠ/ᴼ* (*Figure 1F–K,O–T*). Instead, these guts only contained GFP⁺ precursor cells (*Figure 1F–K,O–T*), suggesting that BMP signaling is also required for intestinal epithelium differentiation.

## BMP signaling is required for ISC self-renewal

The observed reduction of mitotic index in BMP receptor knockdown midguts could be due to reduced stem cell activity or reduced stem cell number. To distinguish these possibilities, we examined the expression of Dl and Su(H)-lacZ, which mark ISC and EB, respectively (*Figure 1A*; *Micchelli and Perrimon, 2006*; *Ohlstein and Spradling, 2006*, *2007*), in adult midguts expressing either *esgᵗˢ>Tkv-RNAi¹⁰⁵⁸³⁴* or *esgᵗˢ>Put-RNAi*. 8 days after adult flies were cultured at 29°C, the number of Dl⁺ cells dropped significantly in Tkv or Put RNAi guts: less than 10% of esg>GFP⁺ precursor cells were Dl⁺ compared with ~50% in control guts, whereas the number of Su(H)-lacZ⁺ cells increased concomitantly (*Figure 2A–G*). In control guts, most pairs of precursor cells contained one Dl⁺ cell and one Su(H)-lacZ⁺ cell (*Figure 2A,D–D″*). By contrast, most pairs of precursor cells deficient for BMP signaling contained two Su(H)-lacZ⁺ and no Dl⁺ cells (*Figure 2B–C,E–F″*). The stem cell loss phenotype caused by inaction of BMP signaling cannot be rescued by blocking apoptosis (*Figure 2—figure supplement 1*), suggesting that BMP signaling does not simply regulate ISC survival. In fact, an increase in the number of Su(H)-lacZ⁺/Su(H)-lacZ⁺ precursor pairs accompanied by a decrease in the number of Dl⁺/Su(H)-lacZ⁺ precursor pairs strongly suggest that ISC loss in Tkv or Put RNAi guts is due to precocious differentiation of ISC daughter cells into EBs.

To determine whether BMP signaling controls ISC/EB fate choice more definitively, we carried out lineage tracing experiments in which the two ISC daughter cells and their descendants were labeled by RFP⁺ (red) and GFP⁺ (green), respectively, following FRT-mediated mitotic recombination (*Figure 2H*). In this system, transgenic RNAi lines or *UAS* transgenes can be expressed in ISCs using *esg-gal4* to assess the effect of inactivation or overexpression of genes of interest on the fate of the two ISC daughters that will generate distinctively labeled twin-spot clones (*Figure 2H,I*). Furthermore, incorporation of *Gal80ᵗˢ* allows temporal control of transgenic gene expression before and after clonal induction by temperature shift (*Figure 2J*). We named this method 'RGT' for RFP and GFP labeled Twin-spot clonal analysis. The scheme for twin-spot experiments involving Put RNAi is shown in *Figure 2J*. 3–5-day-old control or *esgᵗˢ>Put-RNAi* adult flies containing *FRT82 ubi-RFP/FRT82 ubi-GFP* were grown at 29°C for 7 days prior to clone induction by heat shock. After 1-day recovery at 29°C, the flies were raised at 18°C for 4 days prior to analysis. Temperature downshift restores normal BMP signaling, allowing normal lineage differentiation after clonal induction. Consistent with previous reports (*de Navascues et al., 2012*; *O'Brien et al., 2011*), the majority of twin spots (82%; 90/110) from the control guts contained one multi-cellular clone and one single-cell clone that are derived from asymmetric ISC/EB pairs (*Figure 2I,K–K″′,M*, *Figure 2—figure supplement 2*), and only a small fraction of twin spots contained either two multi-cellular clones derived from symmetric ISC/ISC pairs (10%; 11/110) or two single-cell clones derived from symmetric EB/EB pairs (8%; 9/110) (*Figure 2I,M*, *Figure 2—figure supplement 2*). By contrast, the majority of twin spots (80%; 88/110) from *esgᵗˢ>Put-RNAi* expressing guts falls into the symmetric EB/EB class, and the remaining twin spots (20%; 22/110) belongs to the asymmetric ISC/EB class (*Figure 2L–L″′,M*). Thus, loss of BMP signaling alters the outcome of ISC divisions from mostly asymmetric to predominantly symmetric non-self renewing. These results support the notion that BMP signaling regulates ISC/EB fate choice.

## Inactivation of BMP signaling results in multiple defects in midguts

To confirm the results obtained by RNAi, we generated and analyzed MARCM clones deficient for BMP signaling. *putᴾ* (also known as *put¹⁰⁴⁶⁰*) behaves like a genetic null allele (*Ruberte et al., 1995*). 3–5-day-old adult flies were heat shocked to induce GFP⁺ marked *putᴾ* mutant clones and then raised at 25°C for different periods of time prior to analysis. Because of the regional difference of ISC activity in the *Drosophila* midguts (*Buchon et al., 2013*; *Marianes and Spradling, 2013*), we analyzed the ISC

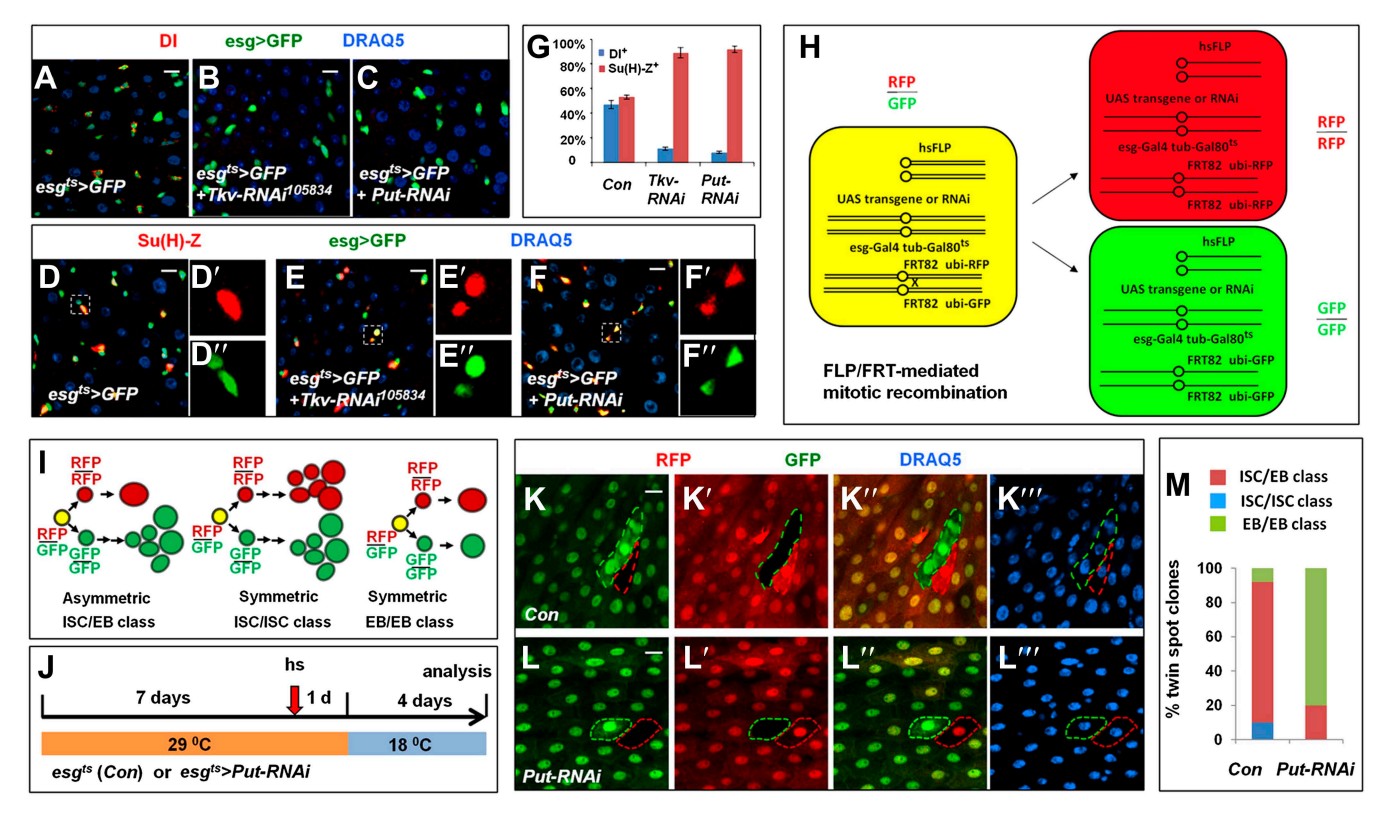

**Figure 2**. BMP signaling is required for ISC self-renewal. (**A–F″**) 3- to 5-day-old adult females expressing *esg*^ts^>GFP (**A**, **D–D″**) or expressing *esg*^ts^>GFP together with *UAS-Tkv-RNAi* (**B**, **E–E″**) or *UAS-Put-RNAi* (**C**, **F–F″**) were shifted to 29°C for 8 days, followed by immunostaining for Dl (red in **A–C**) or Su(H)-lacZ (red in **D–F″**), GFP and DRAQ5 (a nuclear marker). In the control guts, most pairs of precursor cells contain one Dl+ ISC and one Su(H)-lacZ+ EB (**A**, **D–D″**); however, in Tkv or Put RNAi guts, most pairs of precursor cells contain two Su(H)-lacZ+ cells without Dl staining (**B–C**, **E–F″**). (**G**) Percentage of Dl+ or Su(H)-Z+ cells out of GFP+ precursor cells (mean ± SD, n = 10 for each genotype). (**H**) Schematic drawing of an ISC division that produces differentially labeled twin-spot (RFP+ GFP– and RFP– GFP+) through FRT-mediated mitotic recombination. The expression of GFP and RFP is under the control of the *ubiquitin* (*ubi*) promoter. Transgenic overexpression or RNAi through *esg*^ts^ allows determining the effect of gain- or loss-of-function of genes of interest on the outcome of an ISC division. (**I**) Schematic drawings of differentially labeled twin-spot clones generated by FLP/FRT-mediated mitotic recombination of dividing ISCs. (**J**) Scheme for twin-spot experiments involving Put RNAi. 3–5-day-old control or *esg*^ts^>*Put-RNAi* adult flies were grown at 29°C for 7 days before heat shock (hs) to induce clones. After one-day recovery at 29°C, the flies were raised at 18°C for 4 days prior to analysis. (**K–L‴**) Representative twin-spot clones from control and Put RNAi guts. (**M**) Quantification of twin spots of different classes from control and *Put-RNAi* guts: *Con* (n = 110, ISC/EB: 82%, ISC/ISC: 10%, EB/EB: 8%), *Put-RNAi* (n = 110, ISC/EB: 20%, ISC/ISC: 0%, EB/EB: 80%).

The following figure supplements are available for figure 2:

**Figure supplement 1**. Blocking apoptosis does not rescue ISC loss caused by inactivation of BMP signaling.

**Figure supplement 2**. RFP/GFP two-color twin spot clonal analysis.

lineage clones only in the posterior midguts. At 12 days after clone induction (ACI), control ISC lineage clones in the posterior region of midguts contained an average of 8 cells with single Dl+ positive cell and multiple ECs indicated by their large nuclei and Pdm1 staining (*Figure 3A,C,I,J*). By contrast, the majority of *put*^P^ ISC lineage clones contained two cells of small nuclei with no Dl staining but both exhibiting Su(H)-lacZ expression (*Figure 3B,I,J*). Even at an early time point (5 days ACI) whereby most of the ISC lineage clones contain only two cells, 91% (109/120) of the control clones contained one Dl+ cell and one Su(H)-lacZ+ cell whereas 82% (98/120) of the *put*^P^ clones contained two Su(H)-lacZ+ cells (*Figure 3G,H*), suggesting that BMP-signaling-deficient ISC daughters failed to self-renew but instead underwent precocious differentiation into two EBs. Of note, at 12 days ACI, the majority of control clones contained Pdm1+ ECs whereas none of the *put*^P^ clones (0/150)

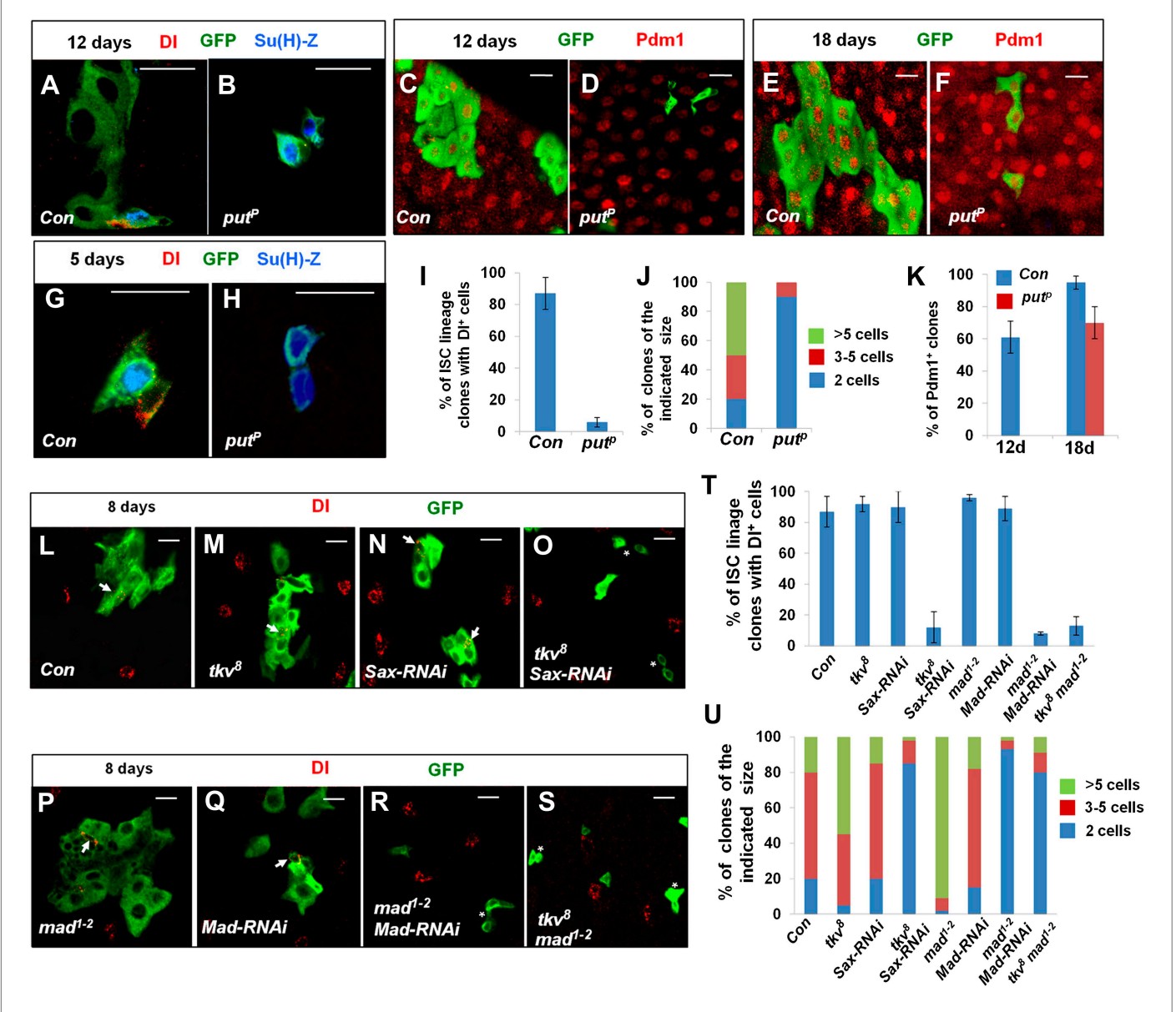

**Figure 3**. Characterization of midgut phenotypes caused by differential inactivation of BMP pathway components. (**A–H**) Midguts containing the control clones (**A**, **C**, **E**, **G**) or *put^P* clones (**B**, **D**, **F**, **H**) were immunostained for Dl (red in **A**, **B**, **G**, **H**) or Pdm1 (red in **C–F**), GFP (green), and Su(H)-lacZ (blue in **A**, **B**, **G**, **H**) at 5 (**G**, **H**), 12 (**A–D**), or 18 (**E**, **F**) days after clone induction (ACI). Control and mutant clones are marked by GFP expression. Control ISC lineage clones usually contain one Dl^+ cell, one or more Su(H)-lacZ^+ cells, and many cells with large nuclei at 12 days ACI. By contrast, most *put* mutant ISC lineage clones contain two cells that are Dl^– but Su(H)-lacZ^+. At 5 days ACI, 91% of control ISC lineage clones contained one Dl^+ cell and one Su(H)-lacZ^+ cell while 82% of *put* mutant ISC lineage clones contained two Su(H)-lacZ^+ cells. (**I**) Quantification of ISC lineage clones containing Dl^+ cells 12 days ACI (mean ± SD, n = 125 for each genotype). (**J**) Quantification of clone size for control (Con) or *put^P* ISC lineage clones 12 days ACI (n = 150 for each genotype). (**K**) Quantification of Pdm1^+ clone frequency for control (Con) and *put^P* ISC lineage clones at 12 or 18 days ACI (mean ± SD, n = 150 for each genotype). (**L–S**) Adult midguts carrying MARCM clones of the indicated genotype were immunostained for Dl and GFP at 8 days ACI. Arrows indicate Dl^+ cells and asterisks in **H** indicate clones without Dl^+ cells. (**T**) Quantification of Dl^+ clone frequency for the indicated genotypes at 8 days ACI (mean ± SD, n = 130 for each genotype). (**U**) Quantification of clone size for the indicated genotypes at 8 days ACI (n = 170 for each genotype).

The following figure supplements are available for figure 3:

**Figure supplement 1**. Tkv and Sax act redundantly in the regulation of ISC self-renewal.

*Figure 3. Continued on next page*

*Figure 3. Continued*

**Figure supplement 2**. Both *tkv$^8$* and *mad$^{1-2}$* mutant clones caused non-cell autonomous ISC overproliferation.

**Figure supplement 3**. Inactivation of BMP signaling in ECs caused ISC proliferation.

exhibited Pdm1 staining (*Figure 3C,D,K*); however, at 18 days ACI, a large fraction of *put$^P$* clones (~70%, n = 95) exhibited Pdm1 staining (*Figure 3E,F,K*). The delayed occurrence of Pdm1$^+$ cells in *put$^P$* clones further argues that BMP signaling is also required for proper ISC lineage differentiation into mature cells.

*tkv$^8$* is a null allele that encodes a truncated protein containing only part of the extracellular domain (*Nellen et al., 1994*). Surprisingly, compared to the control clones, *tkv$^8$* clones exhibited increased clone size and contained one or more Dl$^+$ cells (*Figure 3L,M,T,U*) suggesting that *tkv$^8$* clones over-proliferated. Similar results have been observed by Guo et al (*Guo et al., 2013*). BMP can transduce signal through two type I receptors Tkv and Sax (*Brummel et al., 1994*; *Nellen et al., 1994*), raising a possibility that Sax may support ISC self-renewal in the absence of Tkv. Indeed, we found that *tkv$^8$* clones expressing *Sax-RNAi* behaved similarly to *put$^P$* clones even though Sax-RNAi alone did not exhibit a stem cell loss phenotype (*Figure 3N,O,T*). We noticed that the sequence targeted by *Tkv-RNAi$^{105834}$* contained a region that is conserved between Tkv and Sax (*Figure 3—figure supplement 1A*), suggesting that *Tkv-RNAi$^{105834}$* could inactivate both Tkv and Sax, which may explain why *Tkv-RNAi$^{105834}$* caused ISC loss while *tkv$^8$* did not. Indeed, *Tkv-RNAi$^{40937}$*, which targets a unique region in Tkv did not cause ISC loss but when combined with *Sax-RNAi*, resulted in stem cell loss (*Figure 3—figure supplement 1B–E'*). These results underscore the functional redundancy between Tkv and Sax in the control of ISC self-renewal. Furthermore, they suggest that different degrees of BMP pathway inactivation may result in distinct phenotypes with partial loss of BMP pathway activity causing ISC overproliferation whereas more complete loss of BMP signaling leading to ISC loss.

To further test the idea that different degrees of BMP inactivation have distinct effects on ISC behavior, we generated MARCM clones for a hypomorphic allele of the BMP signal transducer Mad, *mad$^{1-2}$* (Flybase). As expected, *mad$^{1-2}$* clones over-proliferated and behaved like *tkv$^8$* clones (*Figure 3P,T,U*); however, *mad$^{1-2}$* clones expressing *Mad-RNAi* exhibited stem cell loss phenotype (*Figure 3R,T,U*). Strikingly, even though *tkv$^8$* or *mad$^{1-2}$* single mutant clones overproliferated, *tkv$^8$ mad$^{1-2}$* double mutant clones failed to proliferate (*Figure 3U*, *Figure 3—figure supplement 2D*) and exhibited stem cell loss phenotype similar to *put* null mutant clones in the posterior midguts (*Figure 3S,T*).

Interestingly, PH3 staining of midguts containing either *mad$^{1-2}$* or *tkv$^8$* clones revealed increased mitotic index both outside and inside the mutant clones (*Figure 3—figure supplement 2A–C*), suggesting that *mad$^{1-2}$* and *tkv$^8$* clones can exert a non-cell-autonomous effect on the proliferation of neighboring wild type ISCs. In contrast, *tkv$^8$ mad$^{1-2}$* double mutant clones did not exhibit any PH3$^+$ signal, nor did they stimulate the proliferation of neighboring wild type ISCs because no ectopic PH3$^+$ cells were associated with *mad$^{1-2}$ tkv$^8$* double mutant clones in the posterior midguts (*Figure 3U*, *Figure 3—figure supplement 2D*). Similarly, we did not observe any PH3$^+$ cells within or outside of *put* null clones (data not shown). Because *tkv$^8$* and *mad$^{1-2}$* single mutant clones contained many ECs whereas *put* null or *tkv$^8$ mad$^{1-2}$* double mutant clones contained little if any ECs at the time we did PH3 staining, we suspected that the non-cell-autonomous effect of *tkv$^8$* and *mad$^{1-2}$* single mutant clones was due to BMP signaling defects in ECs. In support of this notion, RNAi of Put, Tkv, or Mad in ECs resulted in elevated ISC proliferation (*Figure 3—figure supplement 3*; *Li et al., 2013b*).

## BMP signaling activity is asymmetric in ISC/EB pairs

In *Drosophila*, the BMP signal transducer Mad is phosphorylated upon receptor activation; therefore, the levels of pMad signal are indicative of the levels of BMP pathway activity. By immunostaining with an anti-pMad antibody (*Persson et al., 1998*), we observed high levels of pMad signal in Dl-lacZ$^+$ cells and low levels in Su(H)>GFP$^+$ cells (*Figure 4A–B'*), suggesting that BMP pathway is asymmetrically activated in a pair of ISC/EB cells with ISC transducing higher levels of BMP signaling activity than EB. Consistent with this notion, we found that *dad-lacZ*, which is induced by BMP signaling, exhibits high levels of expression in ISCs and low levels of expression in EBs (*Figure 4C*). However, pMad signals were evenly distributed into two future daughter cells of a dividing ISC marked by PH3$^+$(*Figure 4D*),

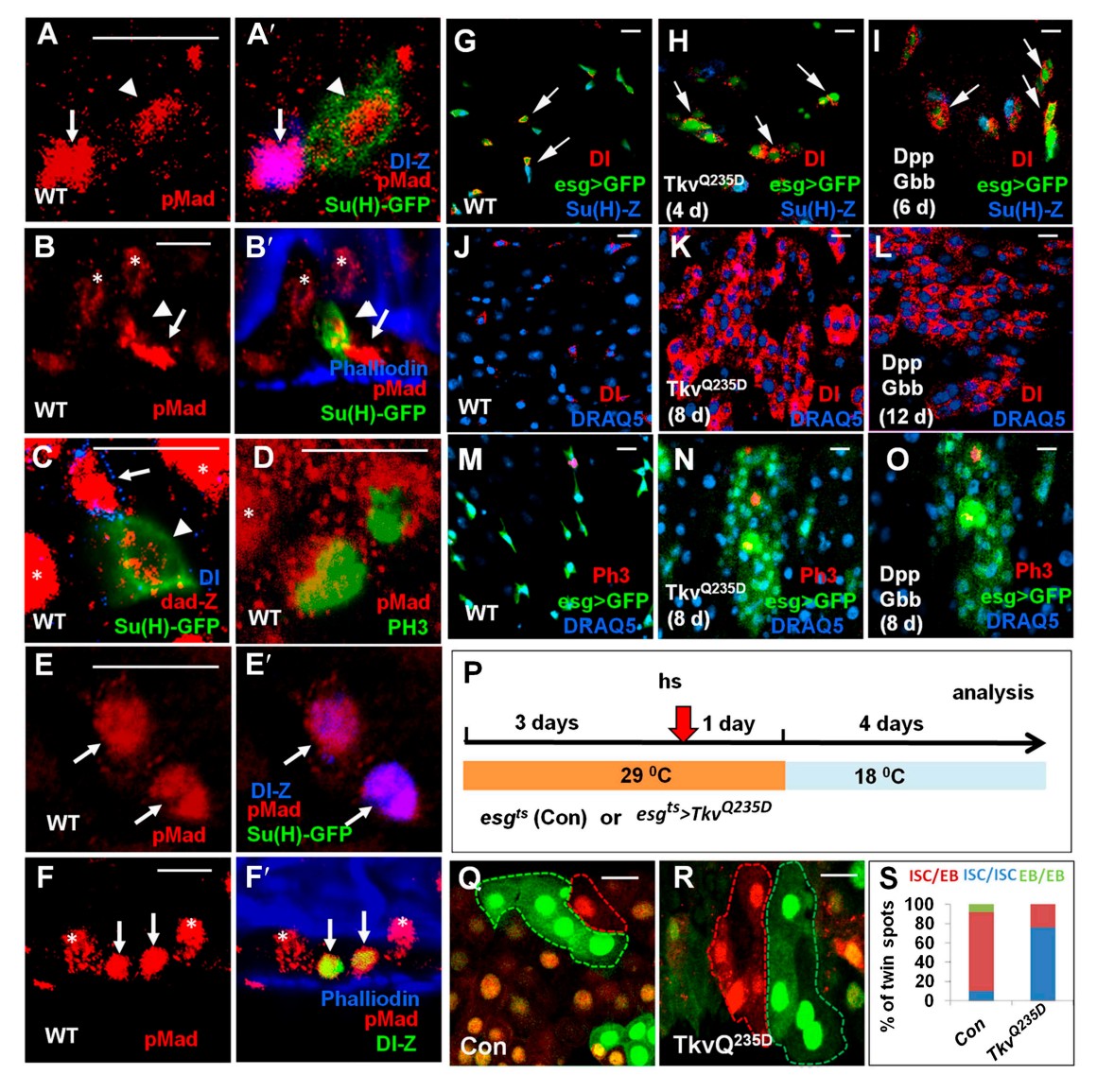

**Figure 4**. Asymmetric BMP signaling regulates ISC self-renewal. (**A–A'**, **C–E'**) High magnification views of wild type adult midguts immunostained for pMad (red in **A–A'**, **D–E'**), dad-lacZ (red in **C**), Dl-lacZ (blue in **A'**, **E'**), Su(H)-GFP (green in **A'**, **C**, **E'**) or PH3 (green in **D**). (**B–B'**, **F–F'**) Sagittal views of wild type adult midguts immunostained for pMad (red), Su(H)-GFP (green in **B'**), Dl-lacZ (green in **F'**), and Phalliodin (blue). Arrows and arrowheads indicate ISCs and EBs, respectively. Asterisks indicate the pMad signals in ECs. (**G–O**) Adult midguts expressing *esg^ts^>GFP* (**G**, **J**, **M**), *esg^ts^>GFP + Tkv^Q235D^* (**H**, **K**, **N**), or *esg^ts^>GFP + Dpp + Gbb* (**I**, **L**, **O**) at 29°C for the indicated time periods were immunostained for Dl (red in **G–L**), PH3 (red in **M–O**), GFP (green), and Su(H)-lacZ or DRAQ5 (blue). (**P**) Scheme for twin-spot experiments involving Tkv^Q235D^ overexpression. 3–5-day-old control or *esg^ts^>Tkv^Q235D^* adult flies were grown at 29°C for 3 days before heat shock (hs) to induce clones. After 1-day recovery at 29°C, the flies were raised at 18°C for 4 days prior to analysis. (**Q–R**) Representative twin-spot clones from control and *esg^ts^>Tkv^Q235D^* guts. (**S**) Quantification of twin spots of different classes from control and *esg^ts^>Tkv^Q235D^* guts: Con (n = 160, ISC/EB: 83%, ISC/ISC: 9%, EB/EB: 8%), Tkv^Q235D^ (n = 190, ISC/EB: 23%, ISC/ISC: 77%, EB/EB: 0%).

The following figure supplements are available for figure 4:

**Figure supplement 1**. Effect of misexpressing Dpp or Gbb alone in precursor cells on ISC self-renewal.

suggesting that asymmetric BMP signaling is unlikely due to asymmetric inheritance of activated pathway components but rather due to asymmetric induction after ISC division. A small fraction (8/105) of ISC division resulted in the production of two Dl+ ISCs that contained equally high levels of pMad staining (***Figure 4E,E'***) and both ISCs tend to lie in close proximity to the BM (***Figure 4F,F'***), suggesting

that they were derived from symmetric cell division (*Goulas et al., 2012*). We also observed strong pMad staining and *dad-lacZ* expression in ECs (indicated by asterisks in *Figure 4B–D, F,F′*), suggesting that BMP signaling is active in differentiated cells.

## Ectopic BMP signaling promotes ISC self-renewal

To determine whether asymmetric BMP signaling plays an instructive role in the regulation of ISC self-renewal, we ectopically activated the pathway in precursor cells by expressing a constitutively active form of Tkv (Tkv$^{Q235D}$) (*Nellen et al., 1996*). Immunostaining with a pMad antibody confirmed that Tkv$^{Q235D}$ induced high levels of BMP pathway activation in precursor cells (data not shown). In control guts, ISCs (Dl$^+$ esg>GFP$^+$) existed in isolation and many of them were accompanied by Su(H)-lacZ$^+$ esg>GFP$^+$ EBs (arrows in *Figure 4G*). 4 days after shifting to 29°C, *esg$^{ts}$>Tkv$^{Q235D}$* guts contained many pairs of Dl$^+$ Su(H)-lacZ$^-$ precursor cells (arrows in *Figure 4H*). Expression of Tkv$^{Q235D}$ for a longer period of time (8 days) resulted in the formation of large clusters of ISC-like cells that contained diving cells (*Figure 4K,N*), suggesting that ectopic BMP signaling promotes ISC self-renewal.

Next, we conducted RGT experiments to confirm that BMP signaling promote ISC fate. The scheme for twin-spot experiments involving Tkv$^{Q235D}$ overexpression is shown in *Figure 4P*. 3–5-day-old control or *esg$^{ts}$>Tkv$^{Q235D}$* adult flies were grown at 29°C for 3 days prior to clone induction by heat shock. After 1-day recovery at 29°C, the flies were raised at 18°C for 4 days to allow lineage differentiation after clonal induction. As shown in *Figure 4Q–S*, midguts expressing *esg$^{ts}$>Tkv$^{Q235D}$* generated symmetric twin clones of the ISC/ISC class at much higher frequency (77%; 146/190) than the control guts (9%; 14/160), suggesting that ectopic BMP signaling activity promotes the symmetric self-renewing outcome of an ISC division. These results demonstrate that excessive BMP signaling favors ISC fate choice.

Dpp and Gbb are the two major BMP ligands in *Drosophila* (*Moustakas and Heldin, 2009*). Coexpression of Dpp and Gbb using *esg$^{ts}$* also resulted in the formation of ISC-like cell clusters similar to ectopic expression of Tkv$^{Q235D}$ (*Figure 4I,L,O*); however, expression of either Dpp or Gbb alone only produced smaller ISC-like cell clusters (*Figure 4—figure supplement 1*; compared with *Figure 4L*). These results suggest that Dpp and Gbb act in concert to promote ISC self-renewal likely by forming a heterodimer (see below) (*Ray and Wharton, 2001*).

## BMP signaling promotes ISC self-renewal by antagonizing Notch

N signaling plays a critical role in balancing ISC self-renewal and differentiation in *Drosophila* midguts. Gain-of-N signaling blocks ISC self-renewal and promotes differentiation whereas loss-of-N signaling leads to excessive ISCs at the expense of EBs (*Figure 5A–A′,F–F′*; *Micchelli and Perrimon, 2006*; *Ohlstein and Spradling, 2006*). BMP signaling could inhibit N pathway activity, thereby blocking ISC differentiation. Alternatively, N signaling could promote EB fate by inhibiting BMP pathway activity. To distinguish these two possibilities, we determined the epistatic relationship between BMP and N signaling. In one set of experiments, we expressed a constitutively active form of N (N$^{ICD}$) (*Struhl et al., 1993*) in precursor cells that also transduced high levels of BMP signaling activity due to the ectopic expression of Tkv$^{Q235D}$ or combined expression of Dpp and Gbb. We found that N activation completely suppressed the formation of ISC-like cell clusters induced by ectopic BMP signaling (*Figure 5D–E′* compared with *Figure 5B–C′*). We also found that there were no GFP$^+$ cells in many areas due to the differentiation of precursor cells into ECs induced by N$^{ICD}$ in *esg$^{ts}$>N$^{ICD}$* or *esg$^{ts}$>N$^{ICD}$ + Tkv$^{Q235D}$* or *esg$^{ts}$>N$^{ICD}$ + Dpp + Gbb* (data not shown), and that the remaining GFP$^+$ precursor cells were all Dl$^-$ (*Figure 5D–E′*). In a reciprocal set of experiments, we inactivated N signaling by expressing *esg$^{ts}$>N-RNAi* in precursor cells in which the BMP signaling was blocked due to the expression of *esg$^{ts}$>Tkv-RNAi$^{105834}$* or *esg$^{ts}$>Put-RNAi*. We found that loss of N signaling restored Dl$^+$ cells in BMP signaling deficient precursor cells (*Figure 5I–J′* compared with *Figure 5G–H′*), suggesting that ISC can form in the absence of BMP signaling as long as N signaling is blocked. Furthermore, pMad staining indicates that most of the ISC-like cells in N RNAi guts exhibited low levels of BMP pathway activity (*Figure 5L–L′* compared with *Figure 5K–K′*; *Figure 5—figure supplement 1*), consistent with N acting downstream of or in parallel with BMP. Taken together, these results suggest that BMP signaling promotes ISC self-renewal by antagonizing N-mediated differentiation.

We noticed that midguts expressing *esg$^{ts}$>Tkv-RNAi$^{105834}$ + N-RNAi* or *esg$^{ts}$>Put-RNAi + N-RNAi* contained smaller clusters of Dl$^+$ cells compared with *esg$^{ts}$>N-RNAi* guts (*Figure 5F–F′*, *Figure 5I–J′*), suggesting that BMP pathway activity is required for the proliferation of N-signaling-deficient ISCs.

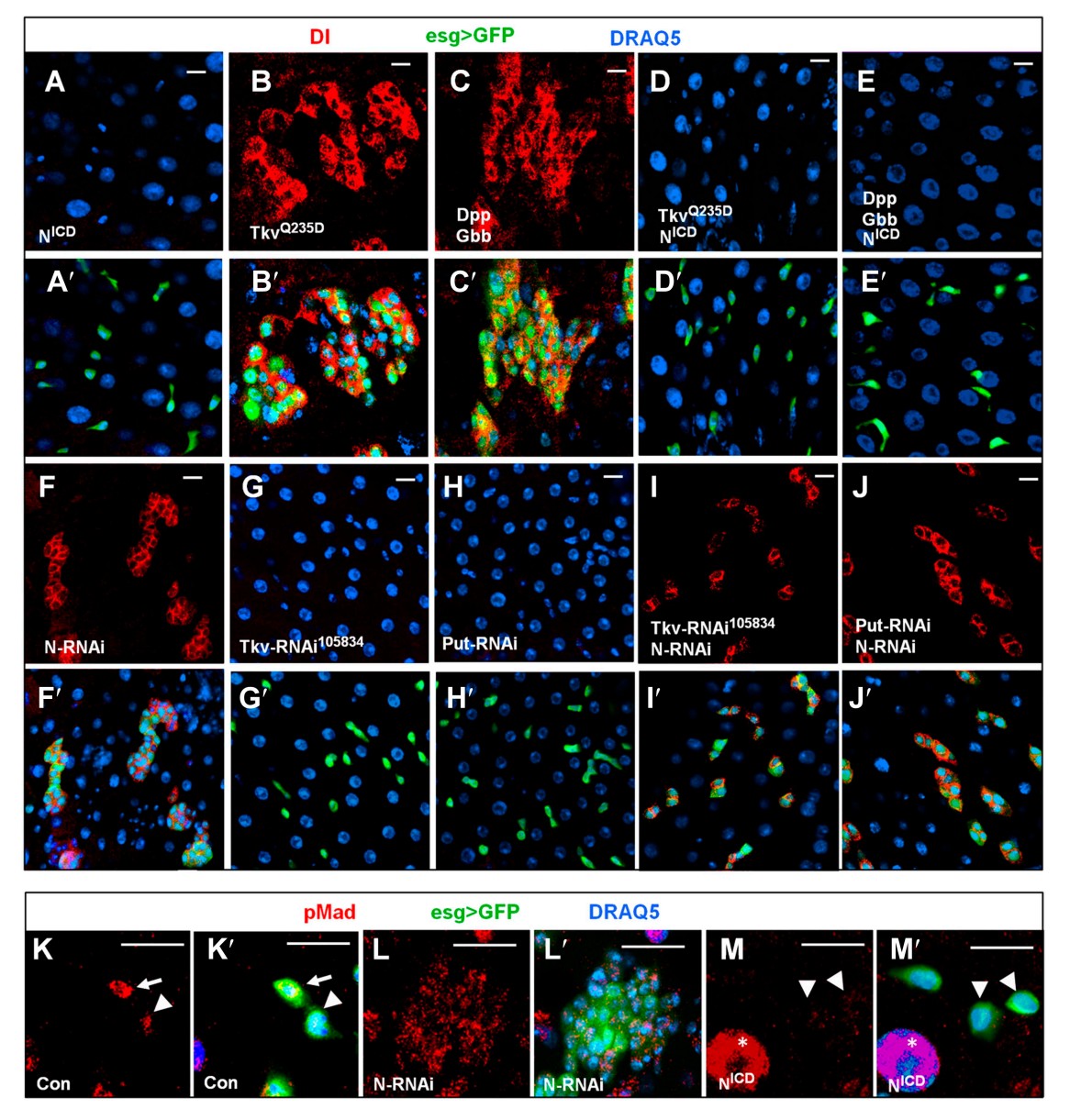

**Figure 5**. BMP signaling promotes ISC self-renewal by antagonizing N. (**A–E'**) Adult midguts expressing N[ICD] (8d) (**A–A'**), Tkv[Q235D] (8d) (**B–B'**), Dpp + Gbb (12d) (**C–C'**), or the indicated combinations of transgenes (**D–E'**) under the control of *esg[ts]* were immunostained for Dl (red), GFP (green) and DRAQ5 (blue). Coexpression of N[ICD] suppressed the excessive Dl+ cells caused by Tkv[Q235D] or Dpp/Gbb misexpression, leading to loss of Dl+ cells similar to expression of N[ICD] alone. (**F–J'**) Adult midguts expressing the indicated RNAi lines under the control of *esg[ts]* for 8 days were immunostained for Dl (red), GFP (green), and DRAQ5 (blue). N RNAi rescued Dl+ cells in midguts expressing *Tkv-RNAi[105834]* or *Put-RNAi*. (**K–M'**) High magnification views of adult midguts expressing *esg[ts]>GFP* (**K–K'**), *esg[ts]>GFP + N-RNAi* (**L–L'**), or *esg[ts]>GFP + N[ICD]* (**M–M'**) and immunostained for pMad (red), GFP (green), and DRAQ5 (blue). Arrows and arrowhead indicate ISC and EB, respectively. Asterisks indicate the pMad signals in ECs.

The following figure supplements are available for figure 5:

**Figure supplement 1**. Integrated pMad levels in control or N knock down precursor cells.

We also noticed that expression of N[ICD] in precursor cells resulted in diminished pMad staining (***Figure 5M–M'*** compared with ***Figure 5K–K'***), suggesting that excessive N signaling can inhibit BMP pathway activity. It is possible that elevated N activity in the presumptive EB can downregulate BMP signaling as a feedback mechanism ('Discussion').

## Dpp and Gbb are produced by ECs

We next sought to determine the source of BMP signals. To our surprise, GFP under the control of a *dpp-Gal4* (*dpp>GFP*) (*Teleman and Cohen, 2000*; *Roy et al., 2011*) was detected in ECs along the anterior-posterior (A-P) axis (*Figure 6A,B,D,D'*) but not in precursor cells or VM (data not shown). However, *dpp>GFP* signals were not uniform along the A-P axis of the midguts but instead, exhibited discrete domains of high-level expression in the posterior (p), middle (m) and anterior (a) regions (*Figure 6A,B*). A similar observation was made by *Li et al. (2013a)*. To confirm *dpp* expression pattern as well as to determine the source of Gbb in midguts, we employed a sensitive RNA in situ hybridization method that allows detection of individual mRNA (*Raj et al., 2008*). Both *dpp* and *gbb* probes could detect endogenous as well as ectopically expressed gene products in wing imaginal discs (*Figure 6— figure supplement 1A–D*). In situ hybridization confirmed that the expression pattern of *dpp>GFP* correlated with that of endogenous *dpp* mRNA (*Figure 6E,E'*, *Figure 6—figure supplement 1E–H*). Furthermore, *dpp* mRNA was not detected in precursor cells or VM (*Figure 6—figure supplement 2A–F'*). Similarly, *gbb* mRNA was detected in ECs but not in precursor cells or VM (*Figure 6F–F'*, *Figure 6— figure supplement 2G–L'*), and high levels of *gbb* were detected in regions that expressed low levels of *dpp* (*Figure 6—figure supplement 1I–L*). This complementary pattern may allow a broad and relatively even distribution of BMP activity along the A/P axis of midguts, as indicated by the *dad-lacZ* expression (*Figure 6C*). Taken together, these results establish ECs as a major source of Dpp and Gbb in adult midguts. Increasing the production of Dpp and Gbb in ECs using the *Myo1A-Gal4/tub-Gal80^{ts}* (*Myo1A^{ts}*) system to express *UAS-Dpp* and *UAS-Gbb* resulted in increased number of Dl+ cells (*Figure 7B*), indicating that paracrine BMP signaling initiated from ECs can promote ISC self-renewal.

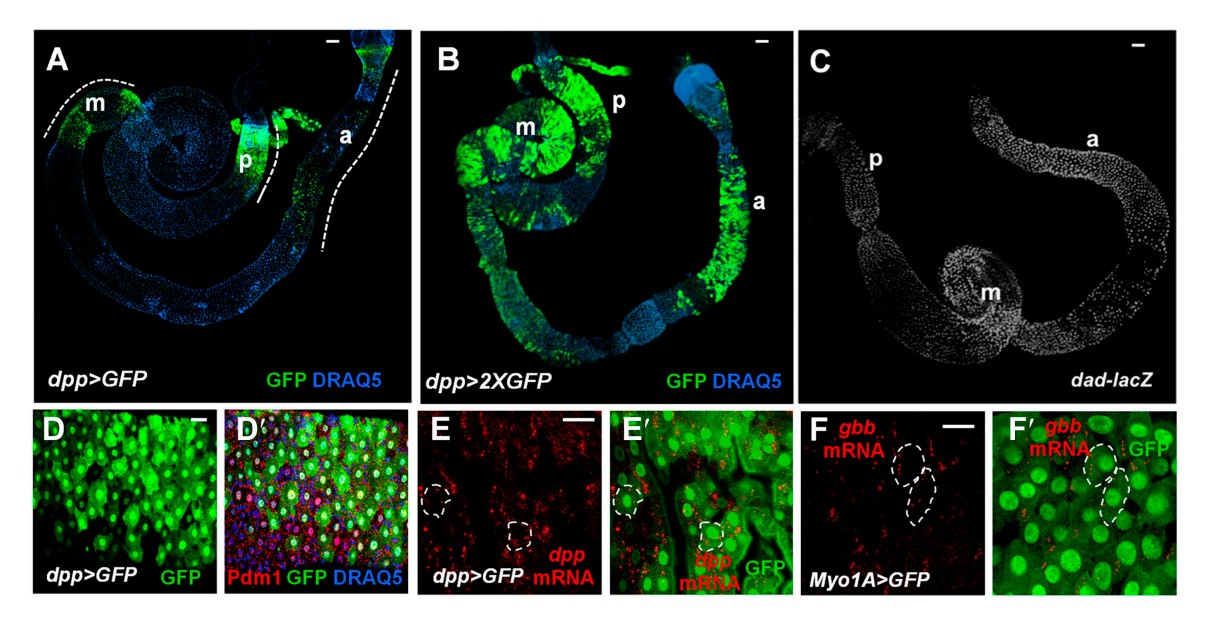

**Figure 6**. Both Dpp and Gbb are expressed in ECs. (**A** and **B**) Low magnification views of adult midguts expressing one (a) or two (b) copies of *UAS-GFP* transgene under the control of *dpp-Gal4* were immunostained for GFP and DRAQ5. *dpp>GFP* is expressed in most of the midgut epithelia with strong expression in the posterior (p), middle (m), and anterior (a) regions. (**C**) Low magnification view of a midgut expressing *dad-lacZ*. (**D–D'**) High magnification view of the posterior region of a *dpp>GFP* expressing midgut immunostained for GFP, Pdm1, and DRAQ5. (**E–E'**) RNAi in situ hybridization of a *dpp>GFP* expressing midgut (posterior region) shows the coincidence of *dpp* mRNA and dpp>GFP signals. *dpp* mRNA signal is detected in the ECs (outlined by dashed line as examples). (**F–F'**) RNA in situ hybridization of midguts expressing *Myo1A>GFP* shows that *gbb* mRNA is detected in ECs. Two ECs are marked by dashed line as examples.

The following figure supplements are available for figure 6:

**Figure supplement 1**. Characterization of *dpp* and *gbb* expression in *Drosophila* by RNA in situ hybridization.

**Figure supplement 2**. *dpp* and *gbb* mRNAs are not detected in precursors or VM.

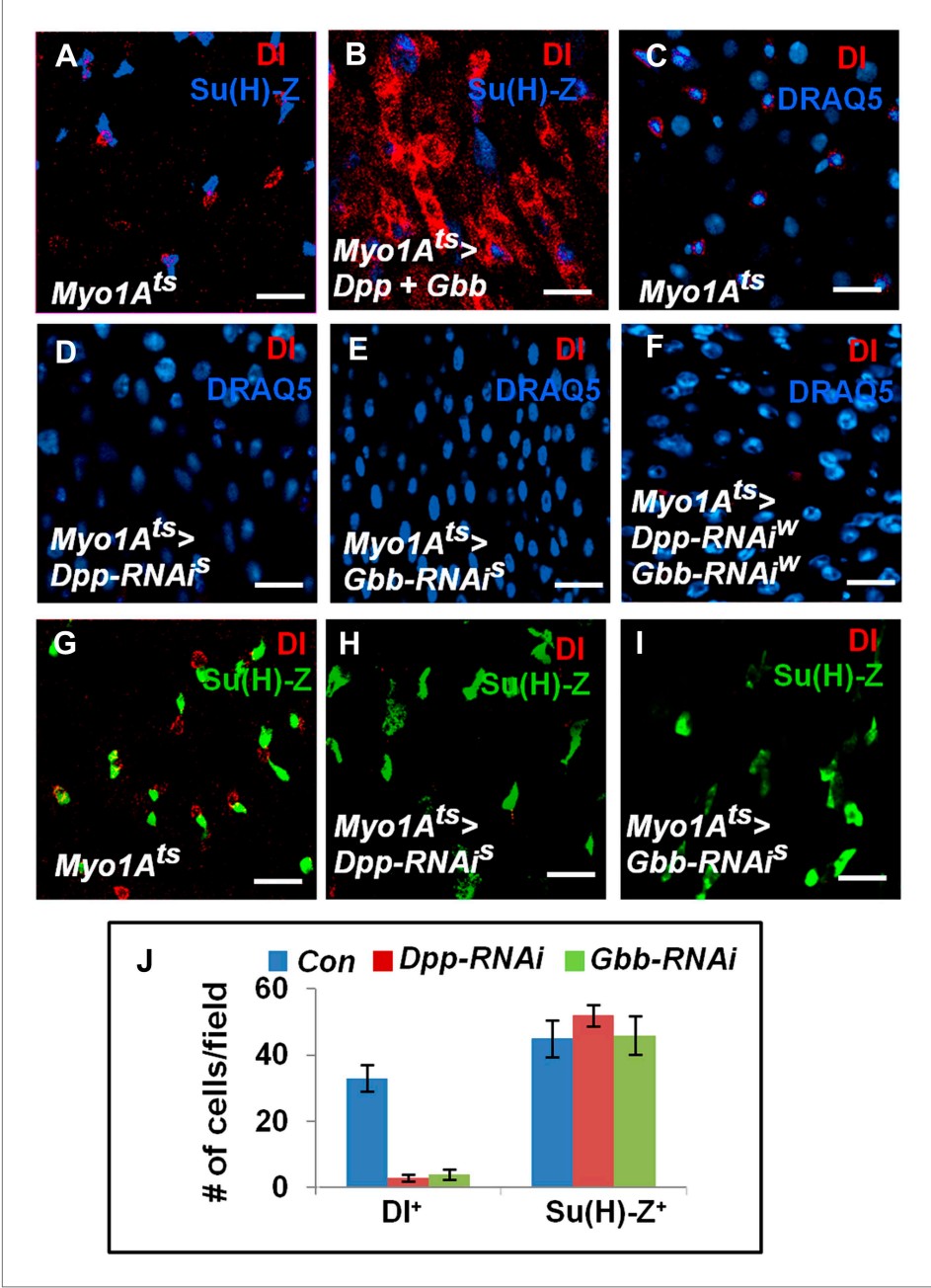

**Figure 7**. EC-derived Dpp and Gbb regulate ISC self-renewal. (**A** and **B**) Control (**A**) or midguts coexpressing both Dpp and Gbb with $Myo1A^{ts}$ (**B**) were immunostained for Dl and Su(H)-lacZ. (**C–I**) Control guts (**C**, **G**), guts expressing strong $Dpp$-$RNAi$ line (**D**, **H**), strong $Gbb$-$RNAi$ line (**E**, **I**), or a combination of weak Dpp- and Gbb-RNAi lines (**F**) were immunostained for Dl (red), Su(H)-lacZ (green), and DRAQ5 (blue). (**J**) Quantification of $Dl^+$ or $Su(H)$-$Z^+$ cell number (mean ± SD, n = 15 for each genotype). Scale bars in **A–C** are 100 µm. Of note, to ensure sufficient knockdown of Dpp or Gbb, two copies of individual RNAi lines were expressed in midguts for 25 days. See 'Materials and methods' the genotypes.

The following figure supplements are available for figure 7:

**Figure supplement 1**. Characterization of Dpp and Gbb RNAi lines.

**Figure supplement 2**. Characterization of Dpp and Gbb knockdown in ECs.

*Figure 7. Continued on next page*

*Figure 7. Continued*

**Figure supplement 3**. Partial loss of BMP in ECs stimulates ISC proliferation.

**Figure supplement 4**. Characterization of Dpp and Gbb in trachea and VM.

## EC-derived Dpp and Gbb are essential for ISC maintenance

If EC-derived Dpp and Gbb serve as niche signals for ISC self-renewal, one would expect that loss of Dpp or Gbb in ECs should lead to stem cell loss. To test this hypothesis, we inactivated Dpp or Gbb in ECs using RNAi. Two Dpp RNAi lines: *UAS-Dpp-RNAi^S* and *UAS-Dpp-RNAi^W*, and two Gbb RNAi lines: *UAS-Gbb-RNAi^S* and *UAS-Gbb-RNAi^W*, were employed. When expressed in wing discs, these RNAi lines caused wing phenotypes indicative of Dpp or Gbb inactivation, and the severity of the wing phenotypes indicated that *UAS-Dpp-RNAi^S* and *UAS-Gbb-RNAi^S* are strong lines whereas *UAS-Dpp-RNAi^W* and *UAS-Gbb-RNAi^W* are weak lines (*Figure 7—figure supplement 1B–E*). This notion was further confirmed by examining the knockdown efficiency using RT-qPCR (*Figure 7—figure supplement 1J*). *Myo1A^ts>Dpp-RNAi* or *Myo1A^ts>Gbb-RNAi* guts were examined for stem cell maintenance 25 days after shifting to 29°C. Of note, two copies of each *UAS-RNAi* line as well as UAS-Dicer2 were coexpressed to increase knockdown efficiency (see 'Materials and methods' for full genotypes). Knockdown of Dpp or Gbb by strong RNAi lines (*Myo1A^ts>Dpp-RNAi^S* or *Myo1A^ts>Gbb-RNAi^S*) resulted in loss of Dl^+ cells (*Figure 7D,E,H–J*; *Figure 7—figure supplement 1I*). pMad staining confirmed that BMP signaling activity was diminished both in precursor cells and ECs in these guts (*Figure 7—figure supplement 2A–C'*). Consistent with the stem cell loss pheno-type, *Myo1A^ts>Dpp-RNAi^S* or *Myo1A^ts>Gbb-RNAi^S* guts exhibited greatly reduced mitotic index in response to injury as compared with controlled guts (*Figure 7—figure supplement 2D–J*). Although knockdown of Dpp or Gbb by weak RNAi lines (*Myo1A^ts>Dpp-RNAi^W* or *Myo1A^ts>Gbb-RNAi^W*) did not significantly affect Dl^+ cell number (*Figure 7—figure supplement 1G–I*), their combined expression (*Myo1A^ts>Dpp-RNAi^W + Gbb-RNAi^W*) resulted in loss of Dl^+ cells (*Figure 7F*; *Figure 7—figure supplement 1I*). Of note, Dpp or Gbb RNAi for shorter period of time (e.g., 10 days) did not cause stem cell loss but instead, resulted in ISC overproliferation due to less complete knockdown (*Figure 7—figure supplement 3*; data not shown).

Recent studies suggested that Dpp is expressed in tracheal cells that contact adult midgut epithe-lium (*Li et al., 2013b*) or in VM (*Guo et al., 2013*). We confirmed that Dpp is expressed in tracheal cells by RNA in situ hybridization (*Figure 7—figure supplement 4A*); however, overexpression of either Dpp alone or in conjunction with Gbb in tracheal cells using *Btl-Gal4 Gal80^ts* (*Btl^ts*) did not increase Dl^+ cell number (*Figure 7—figure supplement 4B–D*). Furthermore, neither Dpp RNAi in trachea (*Btl^ts>Dpp-RNAi^S*) nor in VM (*How^ts>Dpp-RNAi^S*) significantly affected Dl^+ cell number (*Figure 7—figure supplement 4E–H*), which is in contrast to Dpp or Gbb RNAi in ECs. RT-qPCR analysis revealed that *dpp* mRNA was reduced by only ~30% when *dpp* was knocked down in these tissues (*Figure 7—figure supplement 4I*). Furthermore, we did not observed a significant change in pMad staining in *Btl^ts>Dpp-RNAi^S* or *How^ts>Dpp-RNAi^S* midguts (data not shown). Taken together, these results suggest that EC-derived Dpp and Gbb are the major source of BMP that regulates ISC self-renewal.

## BMP is basally enriched

The finding that BMP ligands are produced and required in ECs for ISC self-renewal is counterintuitive because both ISCs and EBs are surrounded by ECs, raising an important question of how the two ISC daughter cells activate different levels of BMP signaling. Interestingly, when expressed in ECs, a GFP-tagged Dpp (Dpp-GFP) was enriched on the basal side of midgut epithelia (*Figure 8B,B'*) whereas a control GFP was uniformly distributed along the apical basal axis (*Figure 8A,A'*). Co-staining with a Golgi marker revealed that Dpp-GFP was enriched on the basal side of the secretary pathway (*Figure 8C*), raising the possibility that Dpp-GFP is preferentially secreted from the basal/basolateral side of ECs. Although a GFP-tagged Gbb (Gbb-GFP) was uniformly distributed along the apical/basal axis when expressed in ECs (*Figure 8—figure supplement 1A,A'*), coexpression of Dpp redistrib-uted Gbb-GFP toward the basal side (*Figure 8—figure supplement 1B–C'*), suggesting that Dpp and Gbb may physically interact, a notion confirmed by coimmunoprecipitation experiments (*Figure 8—figure supplement 2*). Hence, Dpp and Gbb heterodimers may form an apical/basal activity gradient

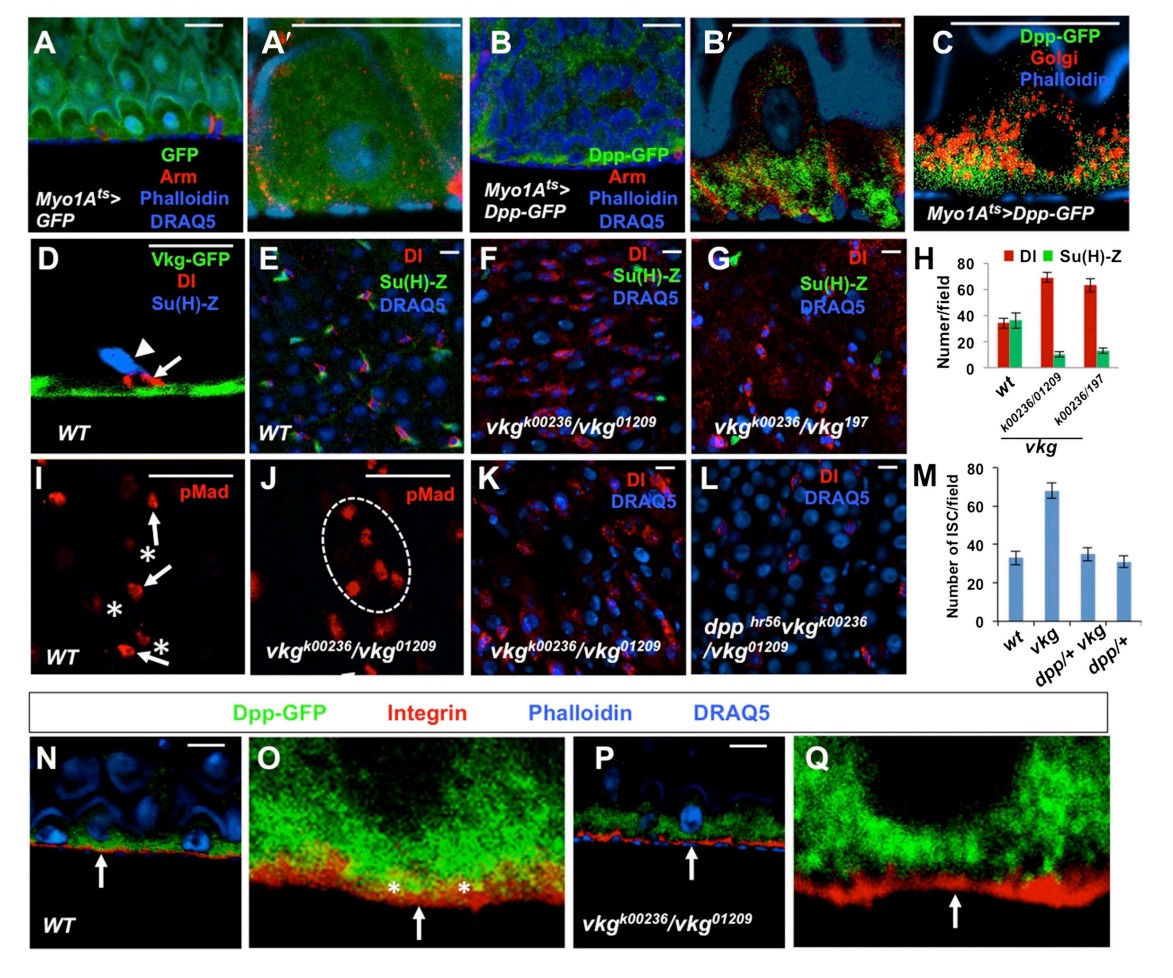

**Figure 8**. Regulation of ISC self-renewal by Vkg and BMP activity gradient. (**A–C**) Low (**A** and **B**) and high (**A'**, **B'**, **C**) magnification sagittal views of adult midguts expressing *Myo1A^ts^>GFP* (**A–A'**) or *Myo1A^ts^>Dpp-GFP* (**B–C**) and immunostained with the indicated antibodies or dyes. Golgi is marked by an anti-Lava lamp antibody. Adult flies expressing Dpp-GFP (or GFP) with *Myo1A^ts^* were raised at 29°C for 5 days, followed by immunostaining. (**D**) Sagittal view of a Vkg-GFP expressing gut immunostained with the indicated antibodies or dyes. Arrows and arrowhead indicate ISC and EB, respectively. (**E–G**, **I**, **J**) Wild type (**E** and **I**) or *vkg* mutant guts (**F**, **G**, **J**) were immunostained for Dl (red in **E–G**), pMad (red in **I** and **J**), Su(H)-lacZ and DRAQ5. In control guts (**I**), precursor cells exhibit high (arrows) and low (asterisks) levels of pMad staining in pairs. *vkg* mutant guts from 10–12-day-old females contained clusters of precursor cells with high levels of pMad signal (outlined in **J**). (**H**) Quantification of Dl+ or Su(H)-Z+ cells in wild type and *vkg* mutant guts (mean ± SD, n = 20 for each genotype). (**K–L**) The ectopic Dl+ phenotype in *vkg* mutant was rescued by *dpp* heterozygosity (*dpp^hr56/+^* at 29°C). (**M**) Quantification of Dl+ cells in midguts of the indicated genotype (mean ± SD, n = 20 for each genotype). *vkg*: *vkg^K00236^/vkg^01209^* ; *dpp^−^/+ vkg*: *dpp^hr56^ vkg^K00236^/vkg^01209^*. (**N–Q**) Low and high magnification views of wild type (**N–O**) or *vkg* mutant (**P–Q**) midguts expressing Dpp-GFP and immunostained for Integrin/Mys, GFP and DRAQ5. Arrows point to the BM in all panels. In wild type guts, Dpp-GFP is enriched at BM as indicated by colocalization of Dpp-GFP and Integrin/Mys signals (yellow in **O**; asterisks), but the colocalization is greatly reduced in *vkg* mutant guts (**Q**).

The following figure supplements are available for figure 8:

**Figure supplement 1**. Dpp regulates Gbb subcellular localization.

**Figure supplement 2**. Dpp physically interacts with Gbb.

**Figure supplement 3**. Integrated pMad levels in precursor cells of control or *vkg* mutant midguts.

**Figure supplement 4**. Dpp localization at the BM is diminished in *vkg* mutant guts.

that allows basally situated ISC daughter cells to transduce higher levels of BMP signal than more apically localized daughter cells.

## BM-localized type IV collagens regulate ISC self-renewal

After an ISC division, the daughter cell that is in close contact with BM usually remains as a stem cell while the more apically localized daughter cell becomes an EB (*Figure 8D*; *Micchelli and Perrimon, 2006*; *Ohlstein and Spradling, 2006*; *Goulas et al., 2012*), raising an interesting possibility that the BM may regulate ISC self-renewal. The type IV collagen encoded by *viking* (*vkg*) is localized at the BM in adult midguts (*Figure 8D*; *Amcheslavsky et al., 2009*). A previous study demonstrated that type IV collagens could physically interact with Dpp to restrict the range of BMP signaling in *Drosophila* ovaries (*Wang et al., 2008*). Interestingly, we found that *vkg* trans-heterozygous midguts (*vkg^{k00236}/vkg^{01209}* and *vkg^{k00236}/vkg^{197}*) contained an excessive number of Dl⁺ cells and greatly reduced number of Su(H)-lacZ⁺ cells (*Figure 8F–H*), suggesting that compromised Vkg activity promotes ISC self-renewal. pMad staining revealed that *vkg* mutant guts contained clusters of precursor cells exhibiting high levels of BMP signaling activity (compare *Figure 8J* with *Figure 8I*, *Figure 8—figure supplement 3*), suggesting that Vkg regulates ISC self-renewal through restricting BMP signaling. As a further support, we found that reducing the dose of Dpp suppressed ectopic ISC formation in *vkg* mutant guts (*Figure 8K–M*). In addition, we found that *vkg* mutant guts retained less Dpp-GFP at the BM compared with control guts (*Figure 8N–Q*, *Figure 8—figure supplement 4*). Hence, the BM in *Drosophila* midguts serves as an integral part of the ISC niche by regulating the extracellular distribution of the niche signal.

## Discussion

Several recent studies revealed that BMP signaling plays several roles in *Drosophila* adult midgut homeostasis (*Guo et al., 2013*; *Li et al., 2013a*, *2013b*). First, peak levels of BMP signaling in the middle region of midguts specify copper cell differentiation (*Guo et al., 2013*; *Li et al., 2013a*). Second, BMP signaling also regulates ISC proliferation because reduction in BMP pathway activity results in ISC overproliferation (*Guo et al., 2013*; *Li et al., 2013b*), which is in line with a growth inhibitory role of BMP signaling in mammalian intestines (*Haramis et al., 2004*; *He et al., 2004*). However, it is controversial whether BMP signaling regulates ISC proliferation cell autonomously or non-cell autonomously (*Guo et al., 2013*; *Li et al., 2013b*). While Li et al. suggested that BMP signaling protects EC integrity and therefore indirectly regulates ISC proliferation; Guo et al. argued that BMP signaling regulates ISC proliferation in a strictly cell-autonomous fashion. The observations that both *tkv^8* and *mad^{1–2}* mutant clones caused excessive proliferation of neighboring wild type ISCs (*Figure 3—figure supplement 2B,C*) and that inactivation of BMP signaling in ECs stimulated ISC proliferation (*Figure 3—figure supplement 3B–E*) clearly support a non-cell autonomous role of BMP signaling. Indeed, inactivation of BMP signaling in ECs stimulated the production of JAK-STAT and EGFR pathway ligands that fuel ISC proliferation (*Li et al., 2013b*) (our own unpublished observation). Nevertheless, it remains possible that BMP signaling could regulate ISC proliferation through both cell autonomous and non cell-autonomous mechanisms, as appear to be the case for Hpo signaling (*Karpowicz et al., 2010*; *Ren et al., 2010*; *Shaw et al., 2010*; *Staley and Irvine, 2010*; *Ren et al., 2013*). Uncertainty also exists regarding how BMP signaling regulates the growth and proliferation of mammalian intestines. While an early work suggested that BMP signaling acts directly on stem/progenitor cells by antagonizing Wnt signaling (*He et al., 2004*), later studies argued that BMP signaling acts on stromal cells to indirectly regulate stem/progenitor cell proliferation (*Kim et al., 2006*; *Auclair et al., 2007*). Future studies are needed to clarify the exact role of BMP signaling in the regulation of ISC proliferation.

In this study, we uncover novel functions of BMP in the regulation of *Drosophila* adult midgut homeostasis, that is, BMP serves as a niche signal to promote ISC self-renewal. In addition, we find that BMP signaling is also required for appropriate lineage differentiation into mature EC and EE. Several lines of evidence suggest that BMP regulates ISC/EB fate choice rather than simply serving as a growth/survival factor for ISC maintenance. First, loss of BMP signaling in precursor cells resulted in a rapid loss of ISC accompanied by an increase in the number of EB. For example, loss of BMP signaling by either *put* RNAi in precursor cells or *put* null ISC lineage clones produced mostly EB/EB pairs as indicated by cell specific markers as well as by two-color lineage tracing experiments

(*Figures 2 and 3*). By contrast, blockage of cell growth/proliferation by inhibiting EGFR pathway or by inactivating dMyc only resulted in a gradual decline in the ISC number; and *Ras* or *dMyc* mutant clones retained the stem cell marker for a long period of time (2–3 weeks) even though they failed to divide (*Xu et al., 2011*; *Ren et al., 2013*). Second, BMP signaling is asymmetric in ISC/EB pairs with the basally located ISCs exhibiting higher levels of BMP signaling activity than the more apically localized EBs. The differential BMP signaling is consistent with BMP pathway activity promoting ISC fate. Third and perhaps the most compelling evidence is that ectopic BMP signaling can promote stem cell fate at the expense of EB fate. For example, gain of BMP signaling either by overexpressing a constitutively active form of Tkv in precursor cells or by misexpressing Dpp and Gbb resulted in the formation of large ISC-like cell clusters (*Figure 4K,L*). Furthermore, the twin-spot lineage tracing experiments confirmed that ectopic BMP signaling favors the symmetric self-renewing (ISC/ISC) outcome of an ISC division.

In seeking for the source of BMP ligands, we were surprised to find that both *dpp* and *gbb* are largely expressed in ECs and their expression patterns are complementary, that is, higher levels of *gbb* mRNA were detected in regions where *dpp* expression is low and vice versa. Low levels of Dpp>GFP expression were detected in ECs along the entire A/P axis (*Figure 6B*) although our RNA in situ hybridization clearly missed low levels of *dpp* mRNA in certain regions. Similarly, our *gbb* RNA in situ probe may have missed low levels of *gbb* expression in the anterior and posterior regions of the midguts. Therefore, it is very likely that Dpp and Gbb are coexpressed in most if not all ECs albeit at different levels in different regions.

A recent study confirmed that *dpp* is expressed in ECs (*Li et al., 2013a*). Although *dpp* expression was also detected in trachea cells (*Figure 7—figure supplement 4A*; *Li et al., 2013b*) as well as in VM (*Guo et al., 2013*), inactivation of Dpp from these tissues did not affect ISC maintenance (*Figure 7—figure supplement 4F–H*). By contrast, inactivation of either Dpp or Gbb in ECs resulted in stem cell loss phenotype whereas increasing the production of Dpp and Gbb in ECs increased stem cell number (*Figure 7*, *Figure 7—figure supplement 1I*). These observations suggest that EC-derived BMP serves as the niche signal for ISC self-renewal although we cannot exclude the possibility that trachea- or VM-derived Dpp could play a minor role. Hence, our study establishes a new paradigm for studying stem cell niche and its regulation because in most other systems, stem cell niches are derived from lineages distinct from the stem cells they support. What then is the advantage of utilizing epithelia as a niche to control ISC self-renewal? We speculate that the employment of midgut epithelia as stem cell niche may provide a mechanism for direct communication between the niche and the environment, allowing the production of niche signal and stem cell number to be regulated in response to various physiological and pathological stimuli. Hence, it would be interesting to explore in the future whether BMP production in ECs is dynamically regulated under various stress conditions where change in the stem cell number has also been observed (*Amcheslavsky et al., 2009*; *Biteau et al., 2008*; *Jiang et al., 2009*; *McLeod et al., 2010*; *O'Brien et al., 2011*).

Our MARCM clone analysis for Tkv shows that loss of Tkv can support the ISC self-renewal, but loss of Sax and Tkv at the same time causes the loss of ISCs. This result suggests that low levels of BMP pathway activity conferred by Sax-Put receptor complex appear to be enough to support ISC self-renewal (although not enough to prevent ISC from overproliferation). On the other hand, our results showed that removal of either Dpp or Gbb resulted in ISC loss, implying that Dpp or Gbb homodimers failed to produce enough BMP activity to support ISC self-renewal. It is not clear why Dpp and Gbb homodimers fail to elicit low levels of BMP pathway activity similar to those transduced by Sax. One possibility is that Dpp and Gbb are produced at much lower levels in the midguts compared to early embryos and imaginal discs so that Dpp and Gbb homodimer concentration may not reach a critical threshold for effective signaling in midgut precursor cells. Second, competition between ECs and ISCs for limited amount of BMP may also restrict the availability of BMP ligands to ISCs. A third possibility is that extracellular Dpp or Gbb homodimers may not be stable in the midguts so that depletion of one ligand might cause concomitant reduction in the levels of the other.

The finding that ECs are the major source of niche signal also raises an important question of how the apical/basal BMP activity gradient is established. Interestingly, we find that Dpp appears to be secreted preferentially from the basal side of ECs, and that Dpp can form a dimer with Gbb to influence its apical/basal distribution. Furthermore, we obtain evidence that BMP signaling range is restricted by the BM-enriched type IV collagen/Vkg. Because type IV collagen/Vkg can physically interact with Dpp (*Wang et al., 2008*), we propose that basal secretion coupled with BM trapping may establish a steep BMP activity gradient so that basally localized ISC daughter cells receive

**Figure 9**. A working model for how BMP regulates ISC self-renewal. Basal/basolateral secretion coupled with basement membrane (BM) trapping sets up an apical-basal BMP activity gradient consisting of Dpp-Gbb heterodimers. Basally localized ISC daughter cells activate higher levels of BMP signaling that promotes ISC self-renewal by antagonizing N. See text for details.

higher levels of BMP than their more apically localized siblings, resulting in differential BMP signaling (*Figure 9*).

Our genetic epistasis experiments suggested that BMP signaling promotes ISC self-renewal by antagonizing N (*Figure 5*). Although it is possible that BMP and N pathways could act in parallel and exert opposing influence on ISC/EB fate choice, we favor a model in which BMP promotes ISC fate by inhibiting N pathway activity because loss of BMP signaling in precursor cells resulted in the ectopic expression of *Su(H)-lacZ*, which is a N pathway activity reporter, whereas gain of BMP signaling suppressed the expression of *Su(H)-lacZ* in precursor cells (*Figures 3 and 4*). Therefore, we propose that differential BMP signaling sets up a difference in the levels of N signaling activity between the two daughter cells after an ISC division. The initial small difference in the N pathway activity between the apical and basal ISC daughter cells is amplified by N feedback regulation (*Figure 9*; *Axelrod, 2010*): N signaling in the apical cell inhibits Dl expression and Dl accumulation in the basal cell further strengthens N signaling in its apical sibling. Intriguingly, we also found that excessive N signaling could block BMP pathway activity, as indicated by the blockage of pMad staining in precursor cells expressing $N^{ICD}$ (*Figure 5M*). We speculate that elevated N pathway activity in the apical ISC daughter cell may attenuate BMP signaling in this cell, forming another feedback mechanism to reinforce EB fate choice. Our model can explain the observations that the absolute levels of BMP pathway activity is not critical for ISC self-renewal since partial loss of BMP signaling did not lead to stem cell loss (*Figure 3*). As long as two ISC daughter cells transduce different levels of BMP signal, the N and BMP signaling feedback loops can amplify the initial small difference, leading to a bistable cell fate choice. A similar mechanism has been postulated to account for the bistable R3/R4 fate determination in the *Drosophila* compound eye, which is regulated by the interplay between Wg/Wnt and N signaling (*Cooper and Bray, 1999*; *Fanto and Mlodzik, 1999*).

Differential BMP signaling might not be the only mechanism responsible for ISC self-renewal. A recent study revealed that aPKC is asymmetrically inherited by apically localized ISC daughter cells and that aPKC promotes N pathway activity (*Goulas et al., 2012*). Therefore, asymmetric segregation of aPKC may dampen BMP signaling response in the apically localized ISC daughter cells, which could contribute to the differential BMP signaling. Future study will determine how BMP signaling inhibits N and how the extrinsic and intrinsic mechanisms are integrated to coordinate ISC self-renewal and differentiation.

## Materials and methods

### *Drosophila* genetics and trangenes

#### Mutant stocks

$put^P$($put^{10460}$) and $tkv^8$ are genetically null alleles; $dpp^{hr56}$ (temperature-sensitive) and $mad^{1-2}$ are hypomorphic alleles (Flybase). $vkg^{K00236}$, $vkg^{197}$, and $Vkg^{01209}$ are hypomorphic alleles (*Wang et al., 2008*). Transgenic RNAi lines: *UAS-Tkv-RNAi*$^{105834}$ (#105834; VDRC); *UAS-Tkv-RNAi*$^{40937}$ (#40937; BL); *UAS-Sax-RNAi* (#42457; VDRC); *UAS-Put-RNAi* (#107071; VDRC); *UAS-mad-RNAi* (#12635; VDRC); *UAS-N-RNAi* (#28981; BL); *UAS-Dpp-RNAi*$^S$ (#33628; BL); *UAS-Dpp-RNAi*$^W$ (#25782; BL); *UAS-Gbb-RNAi*$^S$ (#34898; BL);

*UAS-Gbb-RNAi^W* (*Ballard et al., 2010*); Trangenes: *UAS-Dpp* (#1486; BL); *UAS-Dpp-GFP* (*Entchev et al., 2000*; *Roy et al., 2011*); *UAS-Dpp-HA* (*Shimmi et al., 2005*); *UAS-Gbb* (*Khalsa et al., 1998*); *UAS-Gbb-GFP* (*Nahm et al., 2010*); *UAS-N^ICD* (*Struhl et al., 1993*) *dad-lacZ* (BL# 10305); *UAS-Tkv^Q235D* (*Oh and Irvine, 2011*); *Su(H)-Gal4, Dl-Gal4, Dl-lacZ* and *Su(H)-lacZ* (*Zeng et al., 2010*); *dpp-Gal4* (*Teleman and Cohen, 2000*; *Roy et al., 2011*), *btl-Gal4, how-Gal4, FRT82B ubi-GFP,* and *FRT82B ubi-RFP* (Flybase).

## Clone induction and analysis

Mutant clones for *tkv, put or mad* were generated using the MARCM system (*Lee and Luo, 2001*). Fly stocks were crossed and cultured at 18°C. 5-day-old F1 adults with the appropriate genotypes were subjected to heat shock at 37°C for 1 hr. After clone induction, flies were raised at room temperature for 5, 8, 12, or 18 days before dissection. For experiments involving *tubGal80^ts*, crosses were set up and cultured at 18°C to restrict Gal4 activity. 2 to 3-day-old F1 adult flies were then shifted to 29°C to inactivate Gal80^ts, allowing Gal4 to activate *UAS* transgenes. For twin-spot clonal analysis, 3–5-day-old adult flies were grown at 29°C for 7 days (for Put RNAi experiments) or 3 days (for Tkv^Q235D overexpression experiments) before heat shock at 37°C for 30 min to induce clones. After 1-day recovery at 29°C, the flies were raised at 18°C for 3–4 days. The guts were dissected out and analyzed by confocal microscopy.

## Feeding experiments

In general, 2–3-day-old F1 adult flies were then shifted to 29°C to inactivate Gal80^ts for 8 days, then these adult flies were used for feeding experiments. Flies were cultured in an empty vial containing a piece of 2.5 × 3.75-cm chromatography paper (Fisher, Pittsburgh, PA) wet with 5% sucrose solution as feeding medium. Flies were fed with 5% of DSS (MP Biomedicals, Santa Ana, CA) or 25 µg/ml bleomycin (Sigma, St. Louis, MO) dissolved in 5% sucrose (mock treatment) for 2 days at 29°C.

## Immunostaining

Female flies were used for gut immunostaining in all experiments. The entire gastrointestinal tracts were dissected out and fixed in 1 X PBS plus 8% EM grade paraformaldehyde (Polysciences) for 2 hr. Samples were washed and incubated with primary and secondary antibodies in a solution containing 1 X PBS, 0.5% BSA, and 0.1% Triton X-100. The following primary antibodies were used: mouse anti-Delta (DSHB), 1:100; mouse anti-Pros (DSHB), 1:100; mouse anti-Arm (DSHB), 1:100; mouse anti-integrin βPS/Mys (DSHB), 1:100; rabbit anti-LacZ (MP Biomedicals), 1:1000; rabbit and mouse anti-PH3 (Millipore, Billerica, MA), 1:1000; goat anti-GFP (Abcam, Cambridge, MA), 1:1000; mouse anti-pMad antibody (*Persson et al., 1998*), 1:300; rabbit anti-Pdm1 (gift from Dr Xiaohang Yang, Institute of Molecular and Cell Biology, Singapore), rabbit anti-Lava lamp, 1:300; 1:500; DRAQ5 (Cell Signaling Technology, Danvers, MA), 1:5000; Phalloidin, 1:100. Quantification of immunostaining was performed using ImageJ software.

## RNA in situ hybridization in the adult midguts

RNA fluorescent in situ hybridization (FISH) in the midgut was performed as described (*Raj et al., 2008*). Forty eight 20-mer DNA oligos complementing the coding region of the target genes (*dpp* and *gbb*) were designed and labeled with a fluorophore (http://www.biosearchtech.com/, Petaluma, CA). For RNA in situ hybridization, the midguts were first dissected and fixed in 8% paraformaldehyde at 4°C for overnight, followed by washing with PBS and Triton X-100 (0.1%) for three times (15 min each). The samples were further permeabilized in 70% ethanol at 4°C for overnight. The hybridization was performed according to the online protocol (http://www.biosearchtech.com/stellarisprotocols).

## RT-qPCR

Total RNA was extracted from 10 female guts using RNeasy Plus Mini Kit (#74134; Qiagen, Valencia, CA), and cDNA was synthesized using the iScript cDNA synthesis kit (Bio-Rad, Hercules, CA). RT-qPCR was performed using iQ SYBR Green System (Bio-Rad). Primer sequences used are: 5′-gtgcgaagttt-tacacacaaaga-3′ and 5′-cgccttcagcttctcgtc-3′ (for *dpp*), and 5′-cgctggaactctcgaaataaa-3′ and 5′-ccactt-gcgatagcttcaga-3′ (for *gbb*). RpL11 was used as a normalization control. Relative quantification of mRNA levels was calculated using the comparative $C_T$ method.

## Immunoprecipitation and western blot

For each genotype, 30 Midguts were dissected and mashed in 400 µl lysis buffer: 50 mMTris-HCl (pH 8.0), 100 mM NaCl, 10 mM NaF, 1 mM $Na_3VO_4$, 1% NP40, 10% glycerol, 1.5 mM EDTA (pH 8.0),

protease inhibitor tablets (Roche, IN). 40 µl supernatants were taken out and placed into another tube as the whole cell lysates (WCL). The remaining supernatants were used for IP. Anti-rabbit GFP antibodies and protein A beads were incubated with the lysate for overnight at 4°C, and the beads was washed for three time with the lysis buffer. The immunoprecipitates and WCLs were separated on SDS-PAGE, followed by western blot using anti-GFP and anti-HA antibodies.

## Genotypes for flies in each figure and supplementary figure

**Figure 1**: **B**: *esg^ts*: *w; esg-Gal4 tub-Gal80^ts UAS-GFP/+*, *esg^ts>Tkv-RNAi^105834*: *w; esg-Gal4 tub-Gal80^ts UAS-GFP/UAS-Tkv-RNAi^105834*, *esg^ts>Put-RNAi*: *w; esg-Gal4 tub-Gal80^ts UAS-GFP/UAS-Put-RNAi*. **C–E, L–N**: *w; esg-Gal4 tub-Gal80^ts UAS-GFP/+; UAS-flp, act>CD2>gal4/+*, **F–H, O–Q**: *w; esg-Gal4 tub-Gal80^ts UAS-GFP/UAS-Tkv-RNAi^105834; UAS-flp, act>CD2>gal4/+*, **I–K, R–T**: *w; esg-Gal4 tub-Gal80^ts UAS-GFP/Put-RNAi; UAS-flp, act>CD2>gal4/+*.

**Figure 2**: **A**: *w; esg-Gal4 tub-Gal80^ts UAS-GFP/+*, **B**: *w; esg-Gal4 tub-Gal80^ts UAS-GFP/UAS-Tkv-RNAi^105834*, **C**: *w; esg-Gal4 tub-Gal80^ts UAS-GFP/UAS-Put-RNAi*, **D**: *Su(H)-LacZ; esg-Gal4 tub-Gal80^ts UAS-GFP/+*, **E**: *Su(H)-LacZ; esg-Gal4 tub-Gal80^ts UAS-GFP/UAS-Tkv-RNAi^105834*, **F**: *Su(H)-LacZ; esg-Gal4 tub-Gal80^ts UAS-GFP/UAS-Put-RNAi*, **K**: *yw hsflp; esgGal4 Tub-Gal80^ts; FRT82B ubi-GFP/FRT82B ubi-RFP*, **L**: *yw hsflp; esgGal4 Tub-Gal80^ts/UAS-Put-RNAi; FRT82B ubi-GFP/FRT82B ubi-RFP*.

**Figure 3**: **A, G**: *yw UAS-GFP hsflp/Su(H)-LacZ; tub-Gal4/+; FRT82B tub-Gal80/FRT82B*, **B, H**: *yw UAS-GFP hsflp/Su(H)-LacZ; tub-Gal4/+; FRT82B tub-Gal80/FRT82B put^P*, **C, E**: *yw UAS-GFP hsflp/+; tub-Gal4/+; FRT82B tub-Gal80/FRT82B*, **D, F**: *yw UAS-GFP hsflp/+; tub-Gal4/+; FRT82B tub-Gal80/FRT82B put^P*, **L**: *yw UAS-GFP hsflp; tub-Gal80 FRT40/FRT40; tub-Gal4/+*, **M**: *yw UAS-GFP hsflp; tub-Gal80 FRT40/tkv^8 FRT40; tub-Gal4/+*, **N**: *yw UAS-GFP hsflp; tub-Gal80 FRT40/FRT40; tub-Gal4/UAS-Sax-RNAi*. **O**: *yw UAS-GFP hsflp; tub-Gal80 FRT40/tkv^8 FRT40; tub-Gal4/UAS-Sax-RNAi*, **P**: *yw UAS-GFP hsflp; tub-Gal80 FRT40/mad^1–2 FRT40; tub-Gal4/+*, **Q**: *yw UAS-GFP hsflp; tub-Gal80 FRT40/FRT40; tub-Gal4/UAS-Mad-RNAi*, **R**: *yw UAS-GFP hsflp; tub-Gal80 FRT40/mad^1–2 FRT40; tub-Gal4/UAS-mad-RNAi*, **S**: *yw UAS-GFP hsflp; tub-Gal80 FRT40/tkv^8 mad^1–2 FRT40; tub-Gal4*.

**Figure 4**: **A–B′, E–F′**: *w; Su(H)Gal4 UAS-GFP/+; Dl-LacZ/+*, **C**: *Su(H)Gal4 UAS-GFP/+; dad-LacZ/+*, **G**: *Su(H)-LacZ; esg-Gal4 tub-Gal80^ts UAS-GFP/+*, **H**: *Su(H)-LacZ; esg-Gal4 tub-Gal80^ts UAS-GFP/+; UAS-Tkv^Q235D/+*, **I**: *Su(H)-LacZ; esg-Gal4 tub-Gal80^ts UAS-GFP/+; UAS-Dpp/UAS-Gbb*, **J, M**: *esg-Gal4 tub-Gal80^ts UAS-GFP/+*, **K, N**: *esg-Gal4 tub-Gal80^ts UAS-GFP/+; UAS-Tkv^Q235D/+*, **L, O**: *esg-Gal4 tub-Gal80^ts UAS-GFP/+; UAS-Dpp/UAS-Gbb*. **Q**: *yw hsflp; esgGal4 Tub-Gal80^ts; FRT82B ubi-GFP/FRT82B ubi-RFP*, **R**: *yw hsflp/UAS-Tkv^Q235D; esgGal4 Tub-Gal80^ts/+; FRT82B ubi-GFP/FRT82B ubi-RFP*.

**Figure 5**: **A**: *esg-Gal4 tub-Gal80^ts UAS-GFP/+; UAS-N^ICD/+*, **B**: *esg-Gal4 tub-Gal80^ts UAS-GFP/+; UAS-Tkv^Q235D/+*, **C**: *esg-Gal4 tub-Gal80^ts UAS-GFP/+; UAS-Dpp UAS-Gbb*, **D**: *esg-Gal4 tub-Gal80^ts UAS-GFP/+; UAS-N^ICD/UAS-Tkv^Q235D*, **E**: *esg-Gal4 tub-Gal80^ts UAS-GFP/+; UAS-N^ICD/UAS-Dpp UAS-Gbb*, **F**: *esg-Gal4 tub-Gal80^ts UAS-GFP/+; UAS-N-RNAi/UAS-N-RNAi*, **G**: *esg-Gal4 tub-Gal80^ts UAS-GFP/UAS-Tkv-RNAi^105834*, **H**: *esg-Gal4 tub-Gal80^ts UAS-GFP/UAS-Put-RNAi*, **I**: *esg-Gal4 tub-Gal80^ts UAS-GFP/UAS-Tkv-RNAi^105834; UAS-N-RNAi/UAS-N-RNAi*, **J**: *esg-Gal4 tub-Gal80^ts UAS-GFP/UAS-Put-RNAi; UAS-N-RNAi/UAS-N-RNAi*, **K**: *esg-Gal4 tub-Gal80^ts UAS-GFP*, **L**: *esg-Gal4 tub-Gal80^ts UAS-GFP/+, UAS-N-RNAi/UAS-N-RNAi*. **M**: *esg-Gal4 tub-Gal80^ts UAS-GFP/+; UAS-N^ICD/+*.

**Figure 6**: **A, D, E**: *W; UAS-GFP/+; dppGal4/+*, **B**: *UAS-GFP/UAS-GFP; dppGal4/+*, **F**: *w; UAS-GFP/+; Myo1AGal4/+*.

**Figure 7**: **A**: *Su(H)-LacZ; Myo1AGal4 tub-Gal80^ts UAS-GFP/+*, **B**: *Su(H)-LacZ; Myo1AGal4 tub-Gal80^ts UAS-GFP/+; UAS-Dpp UAS-Gbb/+*, **C**: *Myo1AGal4 tub-Gal80^ts UAS-GFP/UAS-Dicers*, **D**: *Myo1AGal4 tub-Gal80^ts UAS-GFP/UAS-Dicer2; UAS-Dpp-RNAi^S/UAS-Dpp-RNAi^S*, **E**: *Myo1AGal4 tub-Gal80^ts UAS-GFP/UAS-Dicer2; UAS-Gbb-RNAi^S/UAS-Gbb-RNAi^S*, **F**: *Myo1AGal4 tub-Gal80^ts UAS-GFP/UAS-Dicer2; UAS-Dpp-RNAi^W UAS-Gbb-RNAi^W/UAS-Dpp-RNAi^W UAS-Gbb-RNAi^W*, **G**: *Su(H)-LacZ; Myo1AGal4 tub-Gal80^ts UAS-GFP/UAS-Dicer2*, **H**: *Su(H)-LacZ; Myo1AGal4 tub-Gal80^ts UAS-GFP/UAS-Dicer2; UAS-Dpp-RNAi^S/UAS-Dpp-RNAi^S*, **I**: *Su(H)-LacZ; Myo1AGal4 tub-Gal80^ts UAS-GFP/UAS-Dicer2; UAS-Gbb-RNAi^S/UAS-Gbb-RNAi^S*.

**Figure 8**: **A**, *Myo1AGal4 tub-Gal80^ts/UAS-GFP*, **B, C**: *Myo1AGal4 tub-Gal80^ts/+; UAS-Dpp-GFP/+*, **D**: *Su(H)-LacZ; vkg-GFP/+*, **E**: *Su(H)-LacZ*, **F**: *Su(H)-LacZ; vkg^k00236/vkg^01209*, **G**: *Su(H)-LacZ; vkg^k00236/*

*vkg*[197], **J, K**: *vkg*[k00236]/*vkg*[01209], **L**: *vkg*[k00236]*dpp*[hr56]/*vkg*[01209], **N–O**: *Myo1AGal4/+; tub-Gal80*[ts]/*UAS-Dpp-GFP*, **P–Q**: *vkg*[00236]*Myo1AGal4/vkg*[01209]; *tub-Gal80*[ts]/*UAS-Dpp-GFP*.

**Figure 2—figure supplement 1**: **A**: *w; esg-Gal4 tub-Gal80*[ts] *UAS-GFP/+*, **B**: *w; esg-Gal4 tub-Gal80*[ts] *UAS-GFP; UAS-Diap1*, **C**: *w; esg-Gal4 tub-Gal80*[ts] *UAS-GFP/UAS-Put-RNAi*, **D**: *w; esg-Gal4 tub-Gal80*[ts] *UAS-GFP/UAS-Put-RNAi; UAS-Diap1*.

**Figure 2—figure supplement 2**: **A–D**: *yw hsflp; esgGal4 Tub-Gal80*[ts]; *FRT82B ubi-GFP/FRT82B ubi-RFP*.

**Figure 3—figure supplement 1**: **B**: *w; esg-Gal4 tub-Gal80*[ts] *UAS-GFP/+*, **C**: *w; esg-Gal4 tub-Gal80*[ts] *UAS-GFP/UAS-Tkv-RNAi*[40937], **D**: *w; esg-Gal4 tub-Gal80*[ts] *UAS-GFP/UAS-Sax-RNAi*, **E**: *w; esg-Gal4 tub-Gal80*[ts] *UAS-GFP/UAS-Tkv-RNAi*[40937]; *UAS-Sax-RNAi/+*.

**Figure 3—figure supplement 2**: **A**: *yw UAS-GFP hsflp; tub-Gal80 FRT40/FRT40; tub-Gal4/+*, **B**: *yw UAS-GFP hsflp; tub-Gal80 FRT40/tkv*[8] *FRT40; tub-Gal4/+*, **C**: *yw UAS-GFP hsflp; tub-Gal80 FRT40/mad*[1–2] *FRT40; tub-Gal4/+*, **D**: *yw UAS-GFP hsflp; tub-Gal80 FRT40/tkv*[8] *mad*[1–2] *FRT40; tub-Gal4*.

**Figure 3—figure supplement 3**: **A**: *w; myo1A-Gal4 tub-Gal80*[ts] *UAS-GFP*, **B**: *w; myo1A-Gal4 tub-Gal80*[ts] *UAS-GFP/UAS-Dicer2; UAS-Tkv-RNAi*[105834], **C**: *w; myo1A-Gal4 tub-Gal80*[ts] *UAS-GFP/UAS-Put-RNAi*. **D**: *w; myo1A-Gal4 tub-Gal80*[ts] *UAS-GFP/UAS-Dicer2; UAS-mad-RNAi*.

**Figure 4—figure supplement 1**: **A, A'**: *esg-Gal4 tub-Gal80*[ts] *UAS-GFP/+; UAS-Dpp/+*, **B, B'**: *esg-Gal4 tub-Gal80*[ts] *UAS-GFP/+; UAS-Gbb/+*.

**Figure 6—figure supplement 1**: **B**: *MS1096; UAS-dpp*, **D**: *MS1096; UAS-Gbb*, **E–L**: *UAS-GFP/+; dpp-Gal4/+*.

**Figure 6—figure supplement 2**: **A–C'**, **G–I'**: *esg-Gal4/+; UAS-GFP/+*, **D–F'**, **J–L'**: *UAS-GFP/+; how-Gal4/+*.

**Figure 7—figure supplement 1**: **F**: *Myo1AGal4 tub-Gal80*[ts]*UAS-GFP/+*, **G**: *Myo1AGal4 tub-Gal80*[ts]*UAS-GFP/UAS-Dicer2; UAS-Dpp-RNAi*[w]/*UAS-Dpp-RNAi*[w], **H**: *Myo1AGal4 tub-Gal80*[ts]*UAS-GFP/UAS-Dicer2; UAS-Gbb-RNAi*[w]/*UAS-Gbb-RNAi*[w].

**Figure 7—figure supplement 2**: **A, A', D, G**: *Myo1AGal4 tub-Gal80*[ts]*UAS-GFP/+*, **B, B', E, E'**: *Myo1AGal4 tub-Gal80*[ts]*UAS-GFP/UAS-Dicer2; UAS-Dpp-RNAi*[S]/*UAS-Dpp-RNAi*[S], **C, C', F, I**: *Myo1AGal4 tub-Gal80*[ts]*UAS-GFP/UAS-Dicer2; UAS-Gbb-RNAi*[S]/*UAS-Gbb-RNAi*[S].

**Figure 7—figure supplement 3**: **A**: *Myo1AGal4 tub-Gal80*[ts]*UAS-GFP/+*, **B**: *Myo1AGal4 tub-Gal80*[ts]*UAS-GFP/UAS-Dicer2; UAS-Dpp-RNAi*[S]/*UAS-Dpp-RNAi*[S].

**Figure 7—figure supplement 4**: **A**: *btlGal4/+; UAS-GFP*, **B**: *btlGal4/+; tub-Gal80*[ts]*UAS-GFP*, **C**: *btlGal4/+; tub-Gal80*[ts]*UAS-GFP/UAS-Dpp*, **D**: *btlGal4/+; tub-Gal80*[ts]*UAS-GFP/+; UAS-Dpp UAS-Gbb/+*, **E**: *BtlGal4; tub-Gal80*[ts], **F**: *BtlGal4/UAS-Dicer2; UAS-Dpp-RNAi*[S] *tub-Gal80*[ts]/*UAS-Dpp-RNAi*[S], **G**: *tub-Gal80*[ts]*UAS-GFP/UAS-Dicer2; How-Gal4 UAS-Dpp-RNAi*[S]/*UAS-Dpp-RNAi*[S].

**Figure 8—figure supplement 1**: **A**: *Myo1AGal4 tub-Gal80*[ts]; *UAS-Gbb-GFP*, **B**: *Myo1AGal4 tub-Gal80*[ts]; *UAS-Gbb-GFP/UAS-Dpp*, **C**: *Myo1AGal4 tub-Gal80*[ts]; *UAS-Gbb-GFP/UAS-Dpp-HA*.

**Figure 8—figure supplement 4**: **A–A"**, *vkg-GFP*, **B–C**, *Myo1AGal4 tub-Gal80*[ts]/+; *UAS-Dpp-GFP/+*, **D–E**, *vkg*[k00236]*Myo1AGal4/vkg*[01209]; *tub-Gal80*[ts]/*UAS-Dpp-GFP*.

## Acknowledgements

We thank Bing Wang for technical assistance, Drs P ten Dijke, R Cagan, S Newfeld, H Ashe, M O'Connor, T Xie, T Kornberg, S Lee, K Irvine, G Struhl, R Mann, James Skeath, YT Ip, H Jiang, K Wharton, T Schwarz, H Broihier, DSHB, VDRC and Bloomington stock centers for reagents. We thank Drs H Jiang, YT Ip, and G Struhl for discussion.

## Additional information

### Funding

| Funder | Grant reference number | Author |
|---|---|---|
| National Institutes of Health | GM061269 | Jin Jiang |
| Cancer Prevention Research Institute of Texas | RP100561 | Jin Jiang |
| Natural Science Foundation of China | 31328017 | Jin Jiang |

| Funder | Grant reference number | Author |
|---|---|---|
| Welch foundation | I-1603 | Jin Jiang |

The funder had no role in study design, data collection and interpretation, or the decision to submit the work for publication.

## Author contributions

AT, Conception and design, Acquisition of data, Analysis and interpretation of data; JJ, Conception and design, Analysis and interpretation of data, Drafting or revising the article

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
