## [Decision Letter]

Thank you for sending your work entitled “Intestinal epithelium-derived BMP controls stem cell self-renewal in *Drosophila* adult midgut” for consideration at *eLife*. Your article has been favorably evaluated by a Senior editor and 4 reviewers, one of whom is a member of our Board of Reviewing Editors.

The following individuals responsible for the peer review of your submission have agreed to reveal their identity: Utpal Banerjee (Reviewing editor); Xinhua Lin (peer reviewer).

The Reviewing editor and the other reviewers discussed their comments before we reached this decision, and the Reviewing editor has assembled the following comments to help you prepare a revised submission.

Overall, the reviewers have found this work to be important and that the conclusions are well supported by the data. However, please revise your manuscript with the following suggestions in mind.

1) There seems to be considerable disagreement and confusion in the literature over the details of BMP control of cell-fate and renewal in the intestine. In the post review discussion we decided that it is not reasonable to ask you to resolve all the differences experimentally in your manuscript especially since different stocks and conditions are used. However, it is important to point out the differences and the extent to which this manuscript reconciles them. This can be done largely through rewriting the Discussion but perhaps also through small experiments that directly pertain to this manuscript. The primary issues are in the following manuscripts:

A. Zhouhua Li, Yan Zhang, Lili Han, Lai Shi, and Xinhua Lin. Dev. Cell 24, 133-143, 2013.

Conclusion: Trachea-derived Dpp/BMP activates the Dpp signaling in ECs for the maintenance of midgut tissue homeostasis in the adult fly. Loss of BMP signaling in ECs resulted in widespread EC death, which triggered secretion of JAK-STAT and EGFR ligands by dying cells to stimulate ISC proliferation.

A *dpp* source other than the trachea is very unlikely, as systemic knockdown of *dpp* by *tubGal4*^*ts*^ produced virtually identical midgut defects as the trachea knockdowns (data not shown).

B. Zheng Guo, Ian Driver, and Benjamin Ohlstein. J. Cell Biol. 6, 945-961, 2013.

Conclusion: RNAi knockdown of *dpp* in circular muscle or mutant clones of *tkv*^*8*^, *tkv*^*4*^, and *Mad*^*12*^ led to a dramatic increase in PH3-positive cell number in the posterior midgut as compared with sibling controls. Therefore, BMP signaling acts autonomously in ISC to negatively regulate midgut homeostasis.

C. Hongjie Li, Yanyan Qi, and Heinrich Jasper. Cell Reports. 4, 10-18, 2013.

Conclusion: RNAi knockdown of *dpp* in ECs (with NP1-Gal4) or in trachea (btl-Gal4) did not observe a change in ISC proliferation rates (as measured by detecting the number of phosphohistone H3+ cells). Homozygous mutant MARCM clones of *tkv*^*04415*^, *tkva*^*12*^, *Mad*^*1-2*^ and *Mad*^*12*^ didn't cause significant differences compared to wild-type cell clones.

D. Aiguo Tian and Jin Jiang. eLife; Conclusion: Two BMP ligands, Dpp and Gbb, are produced by enterocytes and act in conjunction to promote ISC self-renewal by antagonizing Notch signaling.

Please use the conclusions above only as a guide to compare and contrast the results pointing to the strengths of your manuscript's strengths (and minor weaknesses that can be addressed in future studies).

2) The proposal that low level BMP signaling leads to ISC overproliferation is based on 2 experiments: *tkv*^*8*^ and *mad*^*1,2*^ hypomorph clones, which both develop into bigger than control clones. However additional experiments have also been mentioned that should cause mild inhibition of signaling but that do not elicit the overproliferative response: *Myo*^*ts*^
*dpp RNAi*^w^ or *gbb RNAi*^w^, as well as *sax RNAi* clones all can cause ISC loss in combination with other treatments (indicating that they do reduced signaling) but on their own have no effect and do not induce mild overproliferation. In light of these results it is difficult to accept that low signaling promotes proliferation and perhaps alternative explanations could account for the *tkv*^*8*^ and *mad*^*1,2*^ hypomorphs. Given that this point is rather tangential it can be omitted alongside the *tkv*^*8*^ and *mad*^*1,2*^ data?

3) The data in support of Dpp/Gbb being basally distributed are weak. An intracellular basal (Golgi-associated) accumulation of Dpp makes no prediction on whether it will be secreted basally or basolaterally. It just reflects the steady-state accumulation of Dpp-GFP in the Golgi (most likely the Gbb distribution reflects steady-state accumulation in the ER instead). The signalling pool is the secreted pool and this is the one that needs to be looked at. The authors would have to perform an extracellular staining of dpp-GFP. It would be interesting to compare the extracellular staining of dpp-GFP in wild-type versus *vkg* mutant, as according to their prediction this should lead to a shift in a less basally distributed ligand. However extracellular staining is probably very challenging in the gut. In the absence of stronger data this statement I think needs to be removed. In any case the basal distribution of Viking alone would be sufficient to allow the authors to propose that BMP becomes basally accumulated after secretion, without invoking basal secretion.

4) It does not seem that all the data presented support the existence of a Dpp-Gbb heterodimer. The conclusion from Figure 3 where Tkv and Sax are inactivated singly or together is that these receptors function redundantly. However, this is not consistent with the later evidence for a Dpp-Gbb heterodimer, as the heterodimer would signal synergistically through a Tkv-Put-Sax complex (Shimmi et al., 05, Cell), where removal of either Tkv or Sax would lead to a loss of signaling synergy. It is not clear why removal of Gbb or Dpp singly (Figure 7) gives a similar stem cell loss phenotype (suggesting a lack of potency of say Dpp homodimers), yet removal of Tkv or Sax alone does not generate such a phenotype. Moreover, heterodimerisation of BMPs is thought to occur in the ER (Gray and Mason, 90), which is inconsistent with the conclusion from Figure 6 is that Gbb and Dpp expression is complementary. In contrast, the co-IP and immunostaining suggest Gbb and Dpp interaction. Therefore, I think the authors need to reconcile these different observations.

5) The way the results are described in places does not make the manuscript easily accessible to non-fly readers. For example, while the cartoon in Figure 1 is helpful, a version also showing the overall location of these cells (like in Figure 9) and how this relates to the images shown in Figure 6 would be better for general readers; Figure 1 would have benefitted from a better description of the *esg*^*ts*^ system; and in Figure 4 circling individual cells would be useful.

---

## [Author Response]

*1) There seems to be considerable disagreement and confusion in the literature over the details of BMP control of cell-fate and renewal in the intestine. In the post review discussion we decided that it is not reasonable to ask you to resolve all the differences experimentally in your manuscript especially since different stocks and conditions are used. However, it is important to point out the differences and the extent to which this manuscript reconciles them. This can be done largely through rewriting the Discussion but perhaps also through small experiments that directly pertain to this manuscript. The primary issues are in the following manuscripts*:

*A. Zhouhua Li, Yan Zhang, Lili Han, Lai Shi, and Xinhua Lin. Dev. Cell 24, 133-143, 2013*.

*Conclusion: Trachea-derived Dpp/BMP activates the Dpp signaling in ECs for the maintenance of midgut tissue homeostasis in the adult fly. Loss of BMP signaling in ECs resulted in widespread EC death, which triggered secretion of JAK-STAT and EGFR ligands by dying cells to stimulate ISC proliferation*.

*A* dpp *source other than the trachea is very unlikely, as systemic knockdown of* dpp *by* tubGal4^ts^
*produced virtually identical midgut defects as the trachea knockdowns (data not shown)*.

*B. Zheng Guo, Ian Driver, and Benjamin Ohlstein. J. Cell Biol. 6, 945-961, 2013*.

*Conclusion: RNAi knockdown of* dpp *in circular muscle or mutant clones of tkv*^*8*^*,* tkv^4^*, and* Mad^12^
*led to a dramatic increase in PH3-positive cell number in the posterior midgut as compared with sibling controls. Therefore, BMP signaling acts autonomously in ISC to negatively regulate midgut homeostasis*.

*C. Hongjie Li, Yanyan Qi, and Heinrich Jasper. Cell Reports. 4, 10-18, 2013*.

*Conclusion: RNAi knockdown of* dpp *in ECs (with NP1-Gal4) or in trachea (btl-Gal4) did not observe a change in ISC proliferation rates (as measured by detecting the number of phosphohistone H3+ cells). Homozygous mutant MARCM clones of* tkv^04415^*,* tkva^12^*,* Mad^1-2^
*and* Mad^12^
*didn't cause significant differences compared to wild-type cell clones*.

*D. Aiguo Tian and Jin Jiang. eLife; Conclusion: Two BMP ligands, Dpp and Gbb, are produced by enterocytes and act in conjunction to promote ISC self-renewal by antagonizing Notch signaling*.

*Please use the conclusions above only as a guide to compare and contrast the results pointing to the strengths of your manuscript's strengths (and minor weaknesses that can be addressed in future studies)*.

A) In agreement with Li et al., we also observed elevated ISC proliferation when BMP signaling is inactivated in ECs (Figure 3—figure supplement 3). In addition to detecting *dpp* mRNA in ECs, our RNA in situ probe also detected *dpp* mRNA in tracheal cells (Figure 7—figure supplement 4). The *dpp-Gal4* driver used in our study expressed *UAS-GFP* (*dpp>GFP*) in ECs whereas the *dpp-lacZ* enhancer trap used in Li et al is only expressed in tracheal cells so it seems that each *dpp* reporter only picks up a subset of endogenous *dpp* expression patterns.

However, in contrast to Li et al., we did not observed elevated ISC proliferation when *dpp* was knocked down in tracheal cells (data not shown), neither did we observe stem cell loss phenotype (Figure 7—figure supplement 4). By contrast, we observed ISC loss when *dpp* was knocked down in ECs (Figure 7). Consistent with the notion that ECs are the major source of Dpp, our qRT-PCR analysis revealed that Dpp RNAi in ECs using *Myo1A*^*ts*^ knocked down *dpp* mRNA by ∼70% (Figure 7—figure supplement 1) whereas Dpp RNAi in tracheal cells using *Btl*^*ts*^ only resulted in ∼30% reduction even the same *UAS-Dpp-RNAi* line was used (Figure 7—figure supplement 4). Consistent with ECs being a source of BMP for ISC self-renewal, we found that overexpression of Dpp (+Gbb) in ECs increased ISC number (Figure 7) whereas overexpression of Dpp either alone or together with Gbb in tracheal cells did not (Figure 7—figure supplement 4, D). Therefore, even though tracheal cells express *dpp*, it seems that Dpp produced by these cells may not reach ISCs effectively.

B) In agreement with Guo et al., we also observed increased clone size and PH3 positive cells associated with *tkv*^*8*^ clone as well as *mad*^*1-2*^ clones (Figure 3); however, we found that many PH3 positive cells were located outside *tkv*^*8*^ or *mad*^*1-2*^ clones (Figure 3—figure supplement 2), suggesting that *tkv*^*8*^ and *mad*^*1-2*^ clones stimulated proliferation of neighboring wild type ISCs non-cell-autonomously. In conjunction with the observations that inactivation of BMP signaling in ECs by knocking down individual BMP pathway components also stimulated ISC proliferation (Figure 3—figure supplement 3) (Li et al., Dev Cell 2013), we think that the overproliferation phenotype associated with *tkv* or *mad* clones is largely due to diminished BMP pathway activity in ECs. It is not clear why Guo et al. did not observe such a non-cell-autonomous effect.

Using a 1 kb *dpp* promoter Gal4 fusion to drive *UAS-GFP*, Guo et al. observed dpp>GFP expression in visceral muscles (VM) surrounding the midguts. We used a different *dpp-Gal4* driver, which has been widely used in the literature, to express *UAS-GFP* and found that dpp>GFP is only expressed in ECs. Our RNA in situ hybridization confirmed that *dpp* mRNA was detected in ECs but not in the VM under normal homeostasis. However, it is possible that our RNA in situ hybridization failed to detect low levels of *dpp* mRNA in the VM. Indeed, we found that Dpp RNAi using a VM-selective Gal4 driver (*How*^*ts*^) resulted in a small reduction of *dpp* mRNA; however, we did not observe ISC loss when Dpp was knocked down in the VM (Figure 7—figure supplement 4).

C) In agree with our finding, Li and Jasper also found that *dpp* is expressed in ECs using the same *dpp-Gal4* driver we used as well as RNA in situ hybridization. However, they did not observe a change in ISC proliferation when *dpp* was knocked down in ECs using NP1-Gal4. They also did not observe ISC overproliferation or loss associated with *tkv* or *mad* mutant clones. The difference in the ISC phenotypes between their results and ours is likely due to the different RNAi lines and mutant alleles used and/or different experimental conditions. Even for the same mutant allele (*mad*^*1-2*^ or *mad*^*12*^), different genetic backgrounds might influence the phenotypic outcome. For example, we obtained and analyzed *mad*^*12*^ alleles obtained from three different labs and each exhibited different phenotypes ranging from ISC loss to overproliferation (our unpublished results). This difference also reflects the complex and opposing roles BMP signaling plays in the precursor cells and ECs to regulate ISC self-renewal and proliferation.

*2) The proposal that low level BMP signaling leads to ISC overproliferation is based on 2 experiments:* tkv^8^
*and* mad^1,2^
*hypomorph clones, which both develop into bigger than control clones. However additional experiments have also been mentioned that should cause mild inhibition of signaling but that do not elicit the overproliferative response:* Myo^ts^ dpp RNAi^w^
*or gbb* RNAi^w^*, as well as* sax RNAi *clones all can cause ISC loss in combination with other treatments (indicating that they do reduced signaling) but on their own have no effect and do not induce mild overproliferation. In light of these results it is difficult to accept that low signaling promotes proliferation and perhaps alternative explanations could account for the* tkv^8^
*and* mad^1,2^
*hypomorphs. Given that this point is rather tangential it can be omitted alongside the* tkv^8^
*and* mad^1,2^
*data*?

Our results suggested that ISC loss or overproliferation phenotypes depend on the degree of BMP signaling downregulation and the cell type where BMP signaling is inactivated. Loss but not reduction of BMP signaling in precursor cells resulted in stem cell loss whereas reduction of BMP signaling (likely to a certain threshold) in ECs resulted in ISC proliferation through non-cell-autonomous mechanisms. We think the reason why *tub>Sax-RNAi* and *tub>Mad-RNAi* clones did not over-proliferate by themselves is because *tub-Gal4* driven Sax or Mad RNAi did not cause enough downregulation of BMP signaling. In support of this, we did observe ISC overproliferation when the same *UAS-Mad-RNAi* and *UAS-Sax-RNAi* lines were expressed in ECs using *Myo1A-Gal4*, which is a stronger Gal4 driver than *tub-Gal4* (Figure 3—figure supplement 3; data not shown). With respect to Dpp knock down in ECs, we also observed a similar dosage effect. Although we did not examine whether *Myo1A*^*ts*^*>Dpp-RNAi*^*W*^ or *Myo1A*^*ts*^*>Gbb-RNAi*^*W*^ affect ISC proliferation, we did find that partial knockdown of Dpp by expressing *Myo1A*^*ts*^*>Dpp-RNAi*^*S*^ for shorter period of time (10 days) resulted in ISC overproliferation (we included this in Figure 7—figure supplement 3). Perhaps the most striking observation that supports the dosage effect is that, although both *tkv*^*8*^ and *mad*^*1-2*^ single mutant clones exhibited overproliferation phenotype, *tkv*^*8*^
*mad*^*1-2*^ double mutant clones exhibited ISC loss phenotype (we included this new data in Figure 3).

*3) The data in support of Dpp/Gbb being basally distributed are weak. An intracellular basal (Golgi-associated) accumulation of Dpp makes no prediction on whether it will be secreted basally or basolaterally. It just reflects the steady-state accumulation of Dpp-GFP in the Golgi (most likely the Gbb distribution reflects steady-state accumulation in the ER instead). The signalling pool is the secreted pool and this is the one that needs to be looked at. The authors would have to perform an extracellular staining of dpp-GFP. It would be interesting to compare the extracellular staining of dpp-GFP in wild-type versus* vkg *mutant, as according to their prediction this should lead to a shift in a less basally distributed ligand. However extracellular staining is probably very challenging in the gut. In the absence of stronger data this statement I think needs to be removed. In any case the basal distribution of Viking alone would be sufficient to allow the authors to propose that BMP becomes basally accumulated after secretion, without invoking basal secretion.*

We agree with the reviewers that an intracellular basal (Golgi-associated) accumulation of Dpp makes no prediction on whether it will be secreted basally or basolaterally. We have changed the statement to indicate that the basal intracellular accumulation of Dpp raises a possibility that Dpp might be secreted basally or basolaterally. Because extracellular staining of Dpp-GFP is technically challenging in the gut, we used a basement membrane marker (Integrin) to follow extracellular Dpp-GFP in wild type and *vkg* mutant guts and found that basement membrane localized Dpp-GFP is markedly reduced in *vkg* mutant guts compared with the control guts (Figure 8; Figure 8—figure supplement 4).

*4) It does not seem that all the data presented support the existence of a Dpp-Gbb heterodimer. The conclusion from*
Figure 3
*where Tkv and Sax are inactivated singly or together is that these receptors function redundantly. However, this is not consistent with the later evidence for a Dpp-Gbb heterodimer, as the heterodimer would signal synergistically through a Tkv-Put-Sax complex (Shimmi et al., 05, Cell), where removal of either Tkv or Sax would lead to a loss of signaling synergy. It is not clear why removal of Gbb or Dpp singly (*Figure 7*) gives a similar stem cell loss phenotype (suggesting a lack of potency of say Dpp homodimers), yet removal of Tkv or Sax alone does not generate such a phenotype. Moreover, heterodimerisation of BMPs is thought to occur in the ER (Gray and Mason, 90), which is inconsistent with the conclusion from*
Figure 6
*is that Gbb and Dpp expression is complementary. In contrast, the co-IP and immunostaining suggest Gbb and Dpp interaction. Therefore, I think the authors need to reconcile these different observations*.

The study referred to by the reviewers (58) and other studies indicated that Dpp and another BMP ligand Screw (Scw) act synergistically to confer high levels of BMP signaling activity in early embryos as well as in S2 cells by forming Dpp-Scw heterodimers. In addition, both Tkv and Sax are required for transducing high levels of BMP signaling conferred by the Dpp-Scw heterodimers. Consistent with this, we found that Dpp and Gbb also form heterodimers when coexpressed and both of our gain- and loss-of-function studies indicate that Dpp and Gbb act in concert to regulate ISC self-renewal. We did observe greatly reduced BMP signaling in *tkv* mutant clones as indicated by diminished pMad staining (data not shown); however, low levels of BMP pathway activity conferred by Sax-Put receptor complex appear to be enough to support ISC self-renewal (although not enough to prevent ISC from overproliferation). On the other hand, our results showed that removal of either Dpp or Gbb resulted in ISC loss, implying that Dpp or Gbb homodimers failed to produce enough BMP activity to support ISC self-renewal. It is not clear why Dpp and Gbb homodimers fail to elicit low levels of BMP pathway activity similar to those transduced by Sax. One possibility is that Dpp and Gbb are produced at much lower levels in the midguts compared to early embryos and imaginal discs so that Dpp and Gbb homodimer concentration may not reach a critical threshold for effective signaling in midgut precursor cells. Second, competition between ECs and ISCs for limited amount of BMP may also restrict the availability of BMP ligands to ISCs. A third possibility is that extracellular Dpp or Gbb homodimers may not be stable in the midguts so that depletion of one ligand might cause concomitant reduction in the levels of the other.

In Figure 6, we show that *dpp* and *gbb* expression patterns are complementary, i.e., higher levels of *gbb* mRNA were detected in regions where *dpp* expression is low and vice versa. Low levels of Dpp>GFP expression were detected in ECs along the entire A/P axis (Figure 6) although our RNA in situ hybridization clearly missed low levels of *dpp* mRNA in certain regions. Similarly, our *gbb* RNA in situ probe may have missed low levels of *gbb* expression in the anterior and posterior regions of the midguts. Therefore, it is very likely that Dpp and Gbb are coexpressed in most if not all ECs albeit at different levels in different regions.

*5) The way the results are described in places does not make the manuscript easily accessible to non-fly readers. For example, while the cartoon in*
Figure 1
*is helpful, a version also showing the overall location of these cells (like in*
Figure 9*) and how this relates to the images shown in*
Figure 6
*would be better for general readers;*
Figure 1
*would have benefitted from a better description of the* esg^ts^
*system; and in*
Figure 4
*circling individual cells would be useful.*

We have implemented the reviewers’ suggestions by including a sagittal view of an adult posterior midguts that illustrates the anatomical location of precursor cells and ECs and their apical basal relationship (Figure 1; right panel). We indicated in the legend of Figure 1 that all panels are top views of adult posterior midguts at various magnifications indicated by the scale bars unless indicated otherwise. We also defined *esg*^*ts*^ and *esg*^*ts*^*F/O* systems in the figure legend. We used arrows to indicate individual pairs or clusters of precursor cells in Figure 4.